# A wireless controlled robotic insect with ultrafast untethered running speeds

Zhiwei Liu [1,2,3,4,7], Wencheng Zhan [1,5,7], Xinyi Liu[1], Yangsheng Zhu[1], Mingjing Qi [1,2,3,4], Jiaming Leng[1,2,3,4], Lizhao Wei[1], Shousheng Han[6], Xiaoming Wu[6] ✉ & Xiaojun Yan [1,2,3,4] ✉

Running speed degradation of insect-scale (less than 5 cm) legged micro-robots after carrying payloads has become a bottleneck for microrobots to achieve high untethered locomotion performance. In this work, we present a 2-cm legged microrobot (BHMbot, BeiHang Microrobot) with ultrafast untethered running speeds, which is facilitated by the complementary combination of bouncing length and bouncing frequency in the microrobot's running gait. The untethered BHMbot (2-cm-long, 1760 mg) can achieve a running speed of 17.5 BL s$^{-1}$ and a turning centripetal acceleration of 65.4 BL s$^{-2}$ at a Cost of Transport of 303.7 and a power consumption of 1.77 W. By controlling its two front legs independently, the BHMbot demonstrates various locomotion trajectories including circles, rectangles, letters and irregular paths across obstacles through a wireless control module. Such advancements enable the BHMbot to carry out application attempts including sound signal detection, locomotion inside a turbofan engine and transportation via a quadrotor.

Insects can achieve high-speed linear locomotion and quick turns of small radius[1] to prey or avoid risks. For instance, the maximum running speed of a Californian mite (*Paratarsotomus macropalpis*[2]) can be up to 192.4 BL s$^{-1}$. A tiger beetle from Australia (*Cicindela eburneola*[3]) can also achieve a maximum speed of 171 BL s$^{-1}$. These biological machines motivate the development of insect-scale legged microrobots (dimensions less than 5 cm) for excellent mobility comparable to insects with similar sizes. To fulfill this goal, a variety of insect-scale legged microrobots have been proposed by roboticists and demon-strate satisfactory tethered locomotion performance using external power sources and control electronics[4–6]. However, the tethered movements are highly restricted in confined space and real application scenarios often require high untethered mobility of the microrobot with an onboard power source, control units, and task loads[7–10].

To address this challenge, specific leg gaits similar to legged insects or running mammals have been widely adopted by legged microrobots, and the mimicked gaits fall into two categories, walking gait and running gait[11]. The walking gait utilizes alternate strides of different legs to achieve forward movements in a way similar to cockroaches and a representative example is Harvard HAMR[12–14], which is 4.5 cm in length and driven by eight piezoelectric actuators. The tethered speed of HAMR can reach 11.4 BL s$^{-1}$ and the untethered speed drops to 3.8 BL s$^{-1}$ after carrying onboard power and control units[8]. Due to the complex actuation design, its miniature version HAMR-Jr with a body length of 2.2 cm still faces challenges in achieving untethered movements[15]. As a comparison, the running gait does not require a complex actuation mechanism and it utilizes the bouncing movements of the body at high frequency to mimic the running pos-tures of mammals. The representative examples are several soft microrobots actuated by body deformation[4,16,17]. The presented SEMR[16] microrobot with a body length of 2 cm can achieve an ultrafast tethered running speed (70 BL s$^{-1}$) that even surpasses cheetahs

[1]School of Energy and Power Engineering, Beihang University, Beijing, China. [2]Collaborative Innovation Center of Advanced Aero-Engine, Beijing, China. [3]National Key Laboratory of Science and Technology on Aero-Engine Aero-thermodynamics, Beijing, China. [4]Research Institute of Aero-Engine, Beihang University, Beijing, China. [5]Beijing Key Laboratory of Aero-Engine Structure and Strength, Beijing, China. [6]School of Integrated Circuits, Tsinghua University, Beijing, China. [7]These authors contributed equally: Zhiwei Liu, Wencheng Zhan. ✉e-mail: imewuxm@tsinghua.edu.cn; yanxiaojun@buaa.edu.cn

(30 BL s$^{-1}$). However, when the power and control units are integrated, its untethered speed decreases dramatically to less than 2.5 BL s$^{-1}$ since the body deformation induced bouncing movement is directly affected by the payload. To this date, it remains a great challenge for the legged microrobot to maintain high speed after carrying essential components for untethered locomotion.

In this work, we propose a 2-cm controllable legged microrobot with ultrafast untethered running speeds comparable to that of insects. The bouncing movement is achieved by the periodical impacts between the front legs and the ground using a developed actuation mechanism, and the high untethered mobility is attributed to the complementary combination of bouncing length and high bouncing frequency after carrying payloads. Key advancements of this work include: (i) realization of the running gait based on reciprocating swing motions of front legs with a developed actuation mechanism; (ii) an ultrafast untethered relative running speed at 17.5 BL s$^{-1}$ and a turning centripetal acceleration at 65.4 BL s$^{-2}$ under the dimension limitation of 2 cm; (iii) wireless control along complex trajectories including circles, rectangles, capital letters (BUAA) and irregular paths across obstacles using only two electromagnetic actuators; (iv) application scenarios demonstration of detecting distress signals by carrying a MEMS (Micro-Electro-Mechanical System) microphone, running inside a turbofan engine and collaborating with drones.

## Results

### Design and moving mechanism

The gait analysis in the existing microrobots suggests that the running gait is an effective means of achieving relatively high moving speeds[6,18]. The running gait involves an aerial phase during which the microrobot's legs are momentarily suspended in the air, necessitating a bouncing momentum to propel the microrobot off the ground. This momentum can be achieved not only through body deformation[4,16] but also through the impacts of rigid legs swinging against the ground. The generation of bouncing momentum in an obliquely upward direction requires the impact in an obliquely downward direction upon contact of the swinging legs with the ground. This specific swing motion of the legs forms the fundamental design principle of the BHMbot.

The untethered BHMbot (2.0-cm-long, 1760 mg), which is integrated with an onboard circuit and a lithium battery (3.7 V, 50 mAh), is pictured along a ruler in Fig. 1a. It consists of two electromagnetic actuators, two transmissions which integrate two front legs separately, two rear legs, and the support frames (Fig. 1b). The quantities and masses of all components of the BHMbot are listed in Supplementary Table 1. The electromagnetic actuator with high power density[19] (>200 W kg$^{-1}$) outputs vibratory motions, which consists of a cantilever, a permanent magnet, and a hollow coil. To enhance the performance of the electromagnetic actuator, modeling analysis and parameters optimization have been conducted (Supplementary Note 1 and 2, Supplementary Fig. 1, Supplementary Tables 2 and 3). The planar four-bar linkage transmission acts as a joint connecting the actuator and the front leg (Supplementary Note 3). When an alternating voltage is applied to the coil, the magnet and the cantilever will be excited into vibration by the alternating electromagnetic force from the coil. The reciprocating motion of the magnet is transformed into the swing motion of the actuated leg through the transmission (Fig. 1c). Figure 1d, e show two front legs of a tethered BHMbot (prototype #1, 1.5-cm-long, 370 mg) swinging in phase and out of phase respectively (Supplementary Movie 1). The sizes of several key components are shown in Supplementary Table 4. The fabrication process of the BHMbot is shown in the Methods Section. It is noted that the electromagnetic actuator is adopted for its relatively low operating voltage (<2 V), so a booster module for high voltage is not needed in designing the onboard power and control electronics. From the perspective of kinematics, the presented actuation mechanism design can also be

realized by other linear actuators, such as piezoelectric actuators[20], DEA actuators[21], and electrostatic actuators[22].

To achieve the desired swing motions, the front legs are intentionally designed to be longer than the rear legs to form an upward tilt angle $\theta_0$ of the body[5,23]. This design is crucial for generating obliquely upward bouncing momentum. Initially, the swing angle of the front leg is 0°, as shown in Fig. 1f. When an alternating voltage is applied to the coil, a repulsive electromagnetic force acts on the magnet, causing the front leg to swing forth without resistance due to the existence of the body tilt angle. Subsequently, as the phase of the voltage changes, an attractive force is exerted on the magnet, causing the front leg to swing back. During the backward swing motion, the front leg encounters resistance from the ground. In response, the front leg pushes off the ground, generating an obliquely upward force $F'$. Consequently, the BHMbot gains an obliquely upward momentum, allowing it to bounce off the ground and then land on the ground in a way similar to the locomotion pattern observed in several running mammals, as shown in Fig. 1g. In this situation, the running speed of the BHMbot can be calculated based on the bouncing frequency and the bouncing length during each cycle.

To demonstrate the running gait, a tethered porotype BHMbot (prototype #1) is driven close to its natural frequency (200 Hz) with a running speed of 26 cm s$^{-1}$ (17.5 BL s$^{-1}$), and the forward movement is captured by a high-speed camera (Supplementary Movie 2). The movie indicates that the forward movement of the BHMbot is characterized by a series of continuous bouncing cycles, as shown in Fig. 1g. Figure 1h shows a sequence of optical photos recording one bouncing cycle, which consists of five main phases. In phase 1, both the rear and front legs are in contact with the ground. In phase 2, the rear legs remain in contact with the ground while the front legs swing forth, causing the BHMbot to lower its center of mass (COM). In phase 3, the front legs swing back and exert a backward force against the ground, enabling the BHMbot to acquire an obliquely upward reaction force $F'$. In phase 4, the entire body of the BHMbot bounces off the ground with the front legs swinging forth and back in the air. In the final phase, the BHMbot lands on the ground. Figure 1i provides the measurements of the vertical position of the rear and front feet relative to the ground during one bouncing cycle, with the aerial phase indicated by the blue area. It indicates that a bouncing cycle of the BHMbot normally contains multiple swing cycles of the front legs.

### Tethered locomotion tests and parameters optimization

To determine the proper body length of the BHMbot, we fabricate a series of prototypes (prototype #2.1–2.4) with different body lengths ranging from 10 to 25 mm (see Supplementary Table 4 and Supplementary Note 4) to test the tethered running speeds under varying operating currents and payload masses. Figure 2a, b show the measurements of the relative running speeds of these prototypes without carrying payloads versus the frequency and amplitude of the current. These prototypes can achieve the maximum running speeds when driven near their resonant frequencies. Figure 2a shows the prototypes of 10, 15, 20 and 25 mm achieve maximum speeds of 24.1 BL s$^{-1}$ (24.1 cm s$^{-1}$), 17.5 BL s$^{-1}$ (26.2 cm s$^{-1}$), 14.4 BL s$^{-1}$ (28.8 cm s$^{-1}$), and 11.7 BL s$^{-1}$ (29.2 cm s$^{-1}$), respectively when driven by a current of 0.15 A (Supplementary Fig. 2a and Supplementary Movie 3). The test results indicate that the prototypes with smaller body lengths tend to exhibit higher resonant frequencies and faster relative running speeds near the resonant state.

Figure 2b shows the variation trend of the running speed versus the current amplitude $I_c$. The running speed increases generally with the increase of $I_c$. However, as $I_c$ continues to rise, the speed growth decelerates and even reverses due to the saturation of the electromagnetic driving force. To evaluate the power consumption of the tethered BHMbot, the operating current and voltage are measured simultaneously via a galvanometer when the running speed reaches its

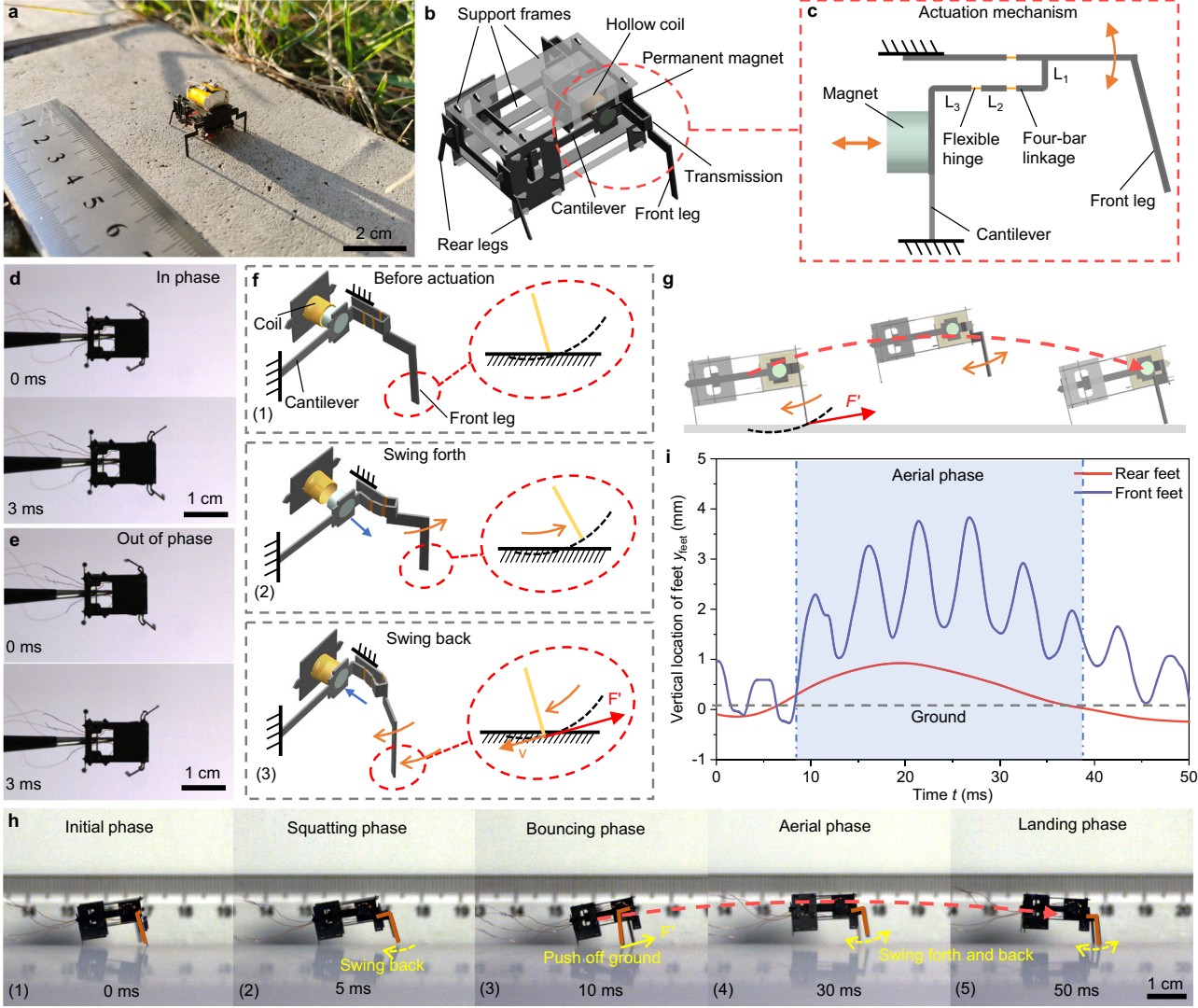

**Fig. 1 | Structural design and moving mechanism of the BHMbot. a** Optical photo of an untethered BHMbot integrated with a lithium battery and an onboard power and control circuit (a ruler is used to show the scale). **b** Three-dimensional model to show the structural details of the BHMbot. **c** Enlarged top view of the actuation mechanism composed of an electromagnetic actuator, a transmission and a front leg. L1, L2 and L3 represent three linkages of the transmission. The reciprocating linear motion of the electromagnetic actuator is transformed into the swing motion of the front leg. **d, e** High-speed images showing the motions of two front legs of the BHMbot (prototype #1) under two in-phase driving signals and two out-of-phase driving signals, respectively. **f** Diagrams illustrating the generation of the bouncing momentum for the BHMbot during an actuation cycle of the front leg. **g** Diagrams showing the bouncing movements of the BHMbot. **h** Series of high-speed images showing one bouncing cycle during the running process of the BHMbot (prototype #1), including initial, squatting, bouncing, aerial, and landing phases. All images are from different parts of Supplementary Movie 2. **i** Measurements of the vertical position of the rear and front feet of the BHMbot relative to the ground during one bouncing cycle. The gray dotted line represents the ground. When all the rear and front feet are off the ground, the BHMbot is in the aerial phase (shown as the blue area).

maximum, as shown in Supplementary Fig. 2b. The power consumption could be estimated as the product of the effective values of the voltage and current signals. Taking prototype #2.2 as an example, the calculated power consumption is 413.6 mW.

Given the significance of payload mass, a series of tests are conducted focusing on the variation trend of the running speed versus the payload mass $m_p$ (Fig. 2c). Throughout the test, $I_c$ remains constant (0.15 A). For the prototype of 10 mm, the running speed drops rapidly with $m_p$ increasing to 600 mg. By contrast, the running speeds of the other three prototypes exhibit an initial increase followed by a decline with respect to $m_p$, indicating that the BHMbot can achieve faster running speed after carrying payloads with a certain mass. In other words, the BHMbot can still realize high-speed untethered locomotion when the total mass of the necessary payloads (such as power and control units) is close to its optimal payload mass. The prototypes of

15, 20 and 25 mm achieve maximum speeds of 29.2 BL s$^{-1}$ (43.8 cm s$^{-1}$), 21.4 BL s$^{-1}$ (42.8 cm s$^{-1}$), and 15.6 BL s$^{-1}$ (39.0 cm s$^{-1}$) when the payload masses are 1200, 2400 and 4800 mg, respectively (Supplementary Movie 3). As anticipated, the larger body lengths can result in higher optimal payload masses. Considering that the estimated mass of the power and control units for generating an alternating current of 0.15 A is about 1.4 g, we select 15 mm as the body length of the BHMbot in the subsequent research for faster untethered running speeds. Supplementary Table 5 contains the maximum running speeds under different payload masses and the corresponding driving frequencies for prototype #2.2.

To investigate the dynamic characteristics of the BHMbot under varying payload masses, we develop a simplified dynamical model of the BHMbot, as shown in Fig. 2d. When the two actuators are driven by the same power channels, the motion of the BHMbot is limited in a

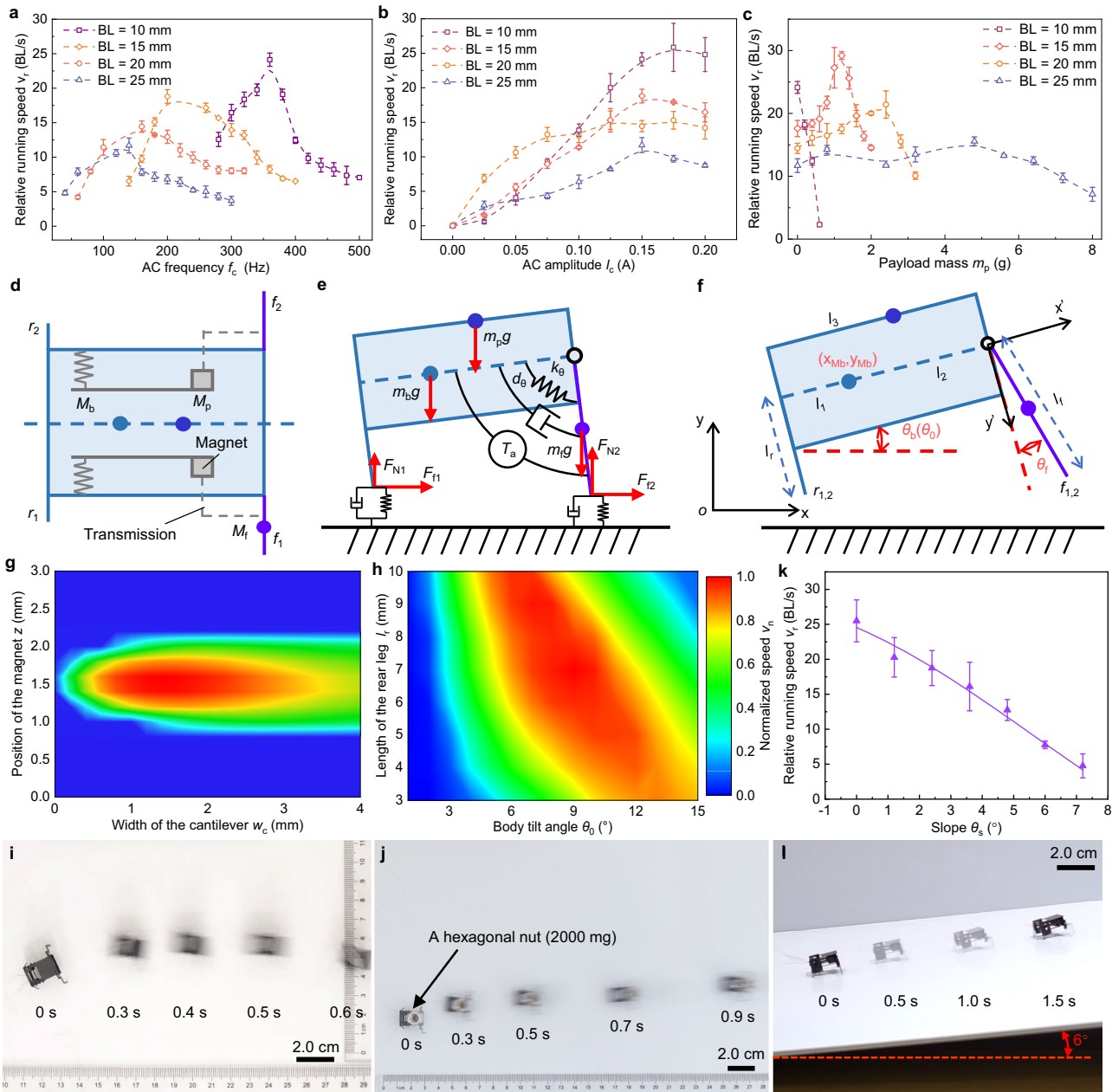

**Fig. 2 | Tethered locomotion tests and parameters optimization.**
**a** Measurements of the relative running speeds of four prototypes with different body lengths (prototype #2.1−2.4, 10, 15, 20 and 25 mm) versus the current frequency, with the current amplitude remaining 0.15 A. **b** Measurements of the relative running speeds of the four prototypes versus current amplitude, with the frequency remaining constant. **c** Measurements of the maximum relative running speeds of the four prototypes versus the payload mass near the resonant state, with the current amplitude remaining 0.15 A. All error bars represent the standard deviation of four measurements. **d** Diagram illustrating the simplified structure of the BHMbot. **e** Simplified planar dynamical model of the BHMbot to analyze the motion characteristics of the BHMbot. **f** Dimensions of the planar dynamical model, including four generalized coordinates used to establish the dynamic equations of the model. **g** Simulation results of normalized relative linear running speed on the paper substrate versus cantilever width $w_c$ and the relative distance $z$ between the

hollow coil and the magnet. **h** Simulation results of normalized relative linear running speed on the paper substrate versus body tilt angle $\theta_0$ and the distance $l_r$ between the rear feet and the COM of the body in the vertical body direction. The color bar represents the magnitude of the normalized running speed, with the red color area indicating the fastest running speed. **i** Optical photo of the tethered BHMbot (prototype #3) using optimized structural parameters moving forward at a maximum speed of 50 cm s⁻¹ (33.3 BL s⁻¹). **j** Optical photo of the tethered BHMbot (prototype #4) moving forward with a relative speed of 25 BL s⁻¹ when carrying a hexagonal nut (2000 mg), which is more than five times its body mass (370 mg). **k** Experimental and simulation results of the relative running speed of the optimized BHMbot (prototype #3) versus the slope angle (from 0° to 7.2°). The error bars represent the standard deviation of four measurements. **l** Optical photo showing prototype #3 moving on a slope of 6° with a maximum speed of 6.5 BL s⁻¹.

plane. In this situation, a planar dynamic model can be utilized to characterize the motion of the BHMbot. As shown in Fig. 2e, the simplified model consists of two rigid bodies $M_b$ and $M_f$, and a lumped mass $M_p$ (Supplementary Note 5). $M_b$ consists of the support frames, the rear legs, the actuators, and the transmissions. $M_f$ and $M_p$ represent

the front legs and the payload, respectively. The pin joint connecting $M_b$ and $M_f$ is simplified as a torsional spring damper ($k_\theta$-$d_\theta$) and is excited by a sinusoidally varying torque source ($T_a$) to simulate the action of the electromagnetic actuators, as shown in Fig. 2e. Four generalized coordinates (Fig. 2f and Supplementary Fig. 3) are selected

to establish the dynamic equations of the BHMbot. The modeling parameters are provided in Supplementary Table 6.

Based on the dynamic model of the BHMbot, the structural parameters are optimized for enhanced running speed and load-carrying capacity (Supplementary Note 6). We select the initial relative distance between the coil and the magnet $z$, the width of the cantilever $w_c$, the initial body tilt angle $\theta_0$, and the length of rear legs $l_r$ (Fig. 2f) as the design parameters to be optimized. To obtain the highest loco-motion mobility, the running speed is selected as the optimization target while keeping the payload mass constant. Figure 2g shows the normalized running speed map of the BHMbot without carrying pay-loads. The color bar represents the magnitude of the normalized running speed, with the red color area indicating the fastest running speed. It is found that $w_c$ near 1.4 mm and $z$ near 1.5 mm are the optimal values for achieving the fastest running speed. Figure 2h shows the normalized running speed map of the BHMbot with respect to the other two variables $\theta_0$ and $l_r$ when keeping $w_c$ and $z$ constant (1.4 mm and 1.5 mm). When $\theta_0$ is near 9° and $l_r$ is near 7 mm, the running speed of the BHMbot reaches its maximum value. When the running speed remains constant, the maximum payload mass can also be optimized through a similar approach.

Utilizing the proposed optimization method, two prototypes (prototype #3 and #4) are designed to demonstrate the enhanced running speeds compared to the initial prototype (prototype #2.2). Prototype #3 (1.5-cm-long, 380 mg) is optimized for faster running speed without carrying payloads. Figure 2i shows prototype #3 runs forward with a maximum speed of 50 cm s$^{-1}$ (33.3 BL s$^{-1}$) (Supplementary Movie 4), surpassing the performance of prototype #2.2 at 17.5 BL s$^{-1}$. Prototype #4 (1.5-cm-long, 380 mg) is designed to demon-strate high-speed locomotion with a payload of 2000 mg, which is the estimated total mass of the lithium battery, power and control elec-tronics, and sensors. Figure 2j shows that prototype #4 exhibits a relative running speed of 25 BL s$^{-1}$ when carrying a payload of 2000 mg (Supplementary Movie 4), which is also higher than that (14.5 BL s$^{-1}$) of prototype #2.2 when the payload mass is 2000 mg. Besides, we have also tested the locomotion performance of the optimized BHMbot (prototype #3) on a series of slopes (0°−7.2°), as shown in Fig. 2k. The prototype #3 can still achieve an average speed of 6.5 BL s$^{-1}$ while climbing a slope of 6° (Fig. 2l).

## Tethered locomotion analysis after carrying payloads

Differing from the existing running microrobots[16,17], the running speed of the BHMbot versus payload mass $m_p$ increases firstly and then declines (Fig. 2c). The optimal payload mass corresponding to the maximum speed is defined as $m_{op}$. Simulation results based on dynamic modeling also indicate that the running speed of the BHMbot keeps increasing until the payload mass exceeds $m_{op}$, which is in accordance with the experimental results (Fig. 3a). This variation trend of the running speed versus $m_p$ can be explained from the perspective of kinematics and dynamics respectively.

Considering the kinematic aspects, the running speed of the BHMbot can be expressed as the product of the bouncing length ($L_{bounce}$) and bouncing frequency ($f_{bounce}$) during the running process:

$$v = f_{bounce} L_{bounce} \qquad (1)$$

Figure 3b, c compare the variation of the vertical position of the COM $y_{Mb}$ and the pitch angle of the rigid body $\theta_b$ of the BHMbot with a payload of 0 mg and 1200 mg ($m_{op}$ of prototype #2.2). When the bare BHMbot (with a payload of 0 mg) is driven close to its resonant fre-quency (200 Hz), both $y_{Mb}$ and $\theta_b$ exhibit irregular variations char-acterized by bouncing cycles with fluctuating periods and amplitudes. In contrast, the BHMbot carrying a payload mass of 1200 mg shows regular variations in $y_{Mb}$ and $\theta_b$ with significantly smaller periods and amplitudes. In other words, although the payload leads to the decrease

of $L_{bounce}$, $f_{bounce}$ increases to counter the decrease of $L_{bounce}$. The complementary combination of $L_{bounce}$ and $f_{bounce}$ contributes to the faster running speed observed in the BHMbot with a payload mass of 1200 mg.

The conclusion drawn from the gait analysis, as observed through a high-speed camera (Supplementary Movie 2), further supports the analysis of the variation trend of the running speed after carrying payloads. Similar to other running microrobots[4,16], there are four main postures observed during the locomotion of the BHMbot, including both-touching, rear-touching, aerial and front-touching postures. Figure 3d, e compare the average duty cycles of the four postures during the locomotion for the bare and load-bearing BHMbot. The bare BHMbot exhibits a significant percentage of aerial duty cycle (>60%, Fig. 3d), whereas the load-bearing BHMbot demonstrates a smaller percentage of aerial duty cycle (<25%, Fig. 3e). The prolonged aerial phase for the bare BHMbot results in multiple actuation cycles of the actuator (Fig. 1h, i) within a bouncing cycle, leading to a lower $f_{bounce}$. As $m_p$ increases, $f_{bounce}$ gradually increases to the actuation frequency and remains constant. Consequently, it is inferred that the initial increase of the running speed is attributed to the dominant increase in $f_{bounce}$ when $m_p$ increases from 0 to $m_{op}$. However, when $m_p$ continues to grow, the increase of $f_{bounce}$ is unable to counter the decrease of $L_{bounce}$, which leads to a decrease of the running speed.

Taking the power flow of the BHMbot into consideration, the power produced by the actuators ($P_a$) can be divided into two parts. The dissipated power ($P_d$) is expended to overcome the damping effects during the swing motion of the front legs. The effective power ($P_e$) which is used to drive the BHMbot moving forward, is regarded as the product of the reaction force from the ground and the speed of the BHMbot:

$$P_e = F_{ff} v \qquad (2)$$

where $F_{ff}$ is the friction force from the ground; $v$ is the speed of the BHMbot. For the bare BHMbot, the front legs remain under actuation during the long aerial phase, resulting in a substantial portion of power being dissipated without contributing to the forward motion of the BHMbot (Fig. 3f). In contrast, when the payload with optimal mass is added, $f_{bounce}$ increases to match the driving frequency of the actua-tors, enabling $P_e$ to reach its maximum value (Fig. 3g). However, as the $m_p$ continues to increase, $P_e$ stops to rise due to the constant $f_{bounce}$, while $P_d$ increases due to the escalating frictional force, which results in the decrease of the running speed.

According to the dynamic analysis, $m_{op}$ is mainly determined by the structural parameters of the BHMbot, including the initial body tilt angle $\theta_0$ (Fig. 2f), the relative location of the COM of the payload mass $l_3$ (Fig. 2f), and the amplitude of the torque applied on the front leg $T_0$ (Fig. 2e), as shown in Fig. 3h and Supplementary Fig. 4. The initial tilt angle $\theta_0$ has a considerable effect on $m_{op}$ and there is an optimal $\theta_0$ for a given payload mass. A detailed discussion about other influencing parameters is provided in Supplementary Note 7.

## Scaling effects analysis

To investigate the scaling effects of the BHMbot further, we conduct a theoretical analysis of scaling effects based on the proposed dynamic model. The scaling process assumes that all geometric parameters of the components of the BHMbot are scaled down in accordance with the ratio of the body length $\phi$ ($\phi = BL_2/BL_1$). The mass of the whole BHMbot will scale down cubically with the size decreasing ($m_2/m_1 = \phi^3$). The torsional stiffness of the front leg also scales down cubically with the size decreasing ($k_{\theta2}/k_{\theta1} = \phi^3$). We select the resonant frequency of the actuation system as an estimate of the resonant fre-quency of the BHMbot $f_r$, and the scaling result is $f_{r2}/f_{r1} = \phi^{-1}$. The calculation of the electromagnetic force has been discussed in Sup-plementary Note 1, and the scaling result is $F_{EM2}/F_{EM1} = \phi^2$. The torque

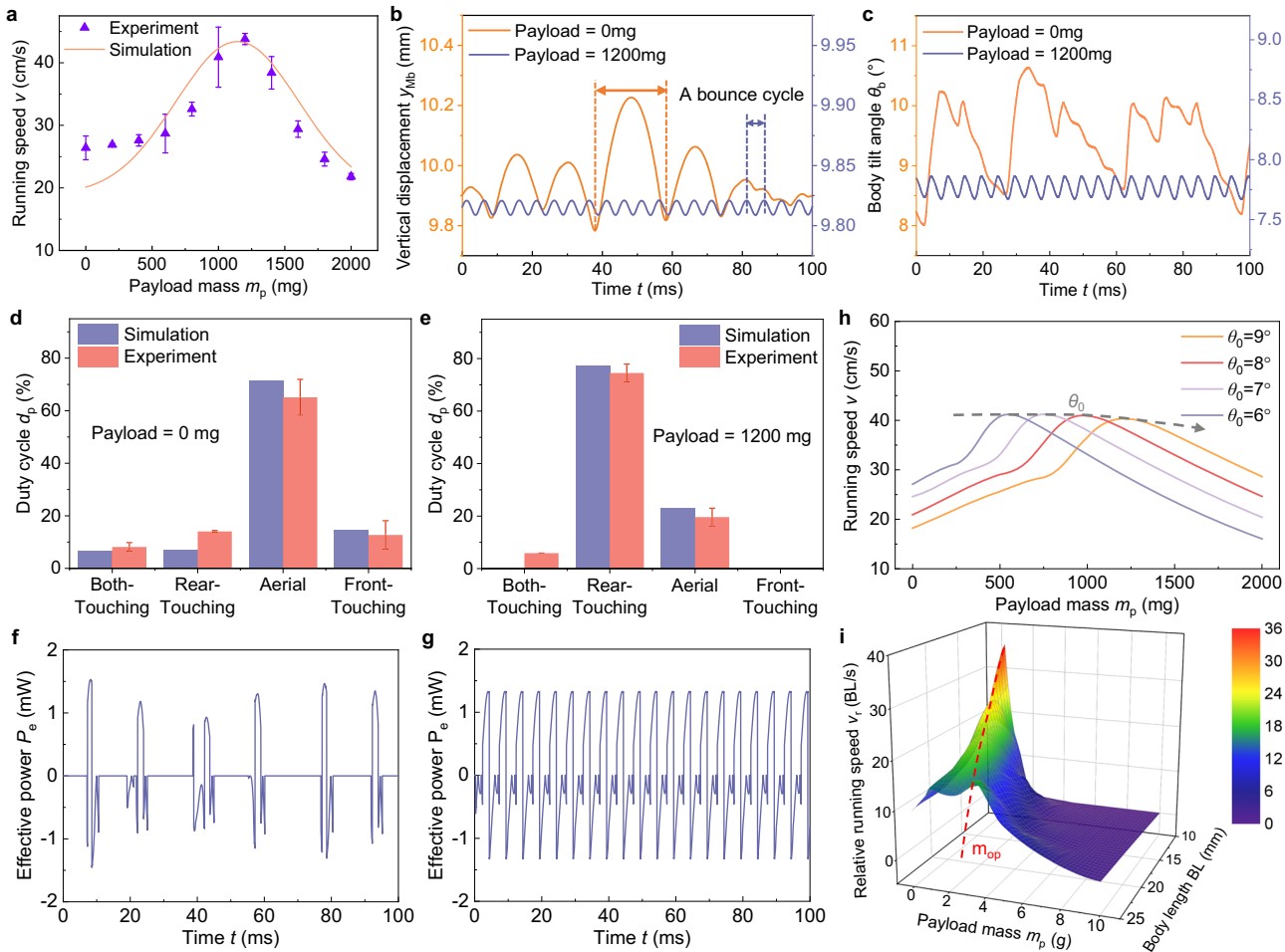

**Fig. 3 | Locomotion performance analysis of the BHMbot after carrying payloads. a** Comparison of the simulation (orange line) and experimental results (purple points) of the maximum running speed versus the payload mass for prototype #2.2. The error bars represent the standard deviation of four measurements. **b** Simulation results of the vertical displacement of the center of mass (COM) of the rigid body $M_b$ for the BHMbot under an actuation frequency of 200 Hz with a payload of 0 mg (orange line) and 1200 mg (blue line). **c** Simulation results of the pitch angle displacement of the rigid body $M_b$ for the BHMbot under an actuation frequency of 200 Hz with a payload of 0 (orange line) mg and 1200 mg (blue line). **d**, **e** Experimental (red bars) and simulation (blue bars) results of the duty cycles of different postures for prototype #2.2 with a payload of 0 mg and 1200 mg. The error bars represent the standard deviation of four measurements. **f**, **g** Simulation

results of the effective power driving the BHMbot forward when the payload mass is 0 mg and 1200 mg respectively. The simulation results are obtained under the same active torque (amplitude: 1.2 mN·m, frequency: 200 Hz). **h** Simulation results of the running speed versus the payload mass for the BHMbot with different initial body tilt angles $\theta_0$ (6°, 7°, 8°, and 9°). **i** Simulation results of the scaling performance of the BHMbot (the maximum relative running speed under varying payload mass), with the body length ranging from 10 mm to 25 mm. The color bar represents the relative running speed and the red color represents the maximum (36 BL s$^{-1}$). In the scaling process, it is assumed that all the geometric parameters of the components of the BHMbot are scaled down based on the ratio of the body length $\phi$.

applied on the front leg is determined by the electromagnetic force and the transmission ratio of the transmission, and the scaling result is $T_{a2}/T_{a1} = \phi^3$.

Utilizing the dynamic model, we can obtain the numerical solutions of the maximum relative running speed $v_{max}$ of the bare BHMbot, the optimal payload mass $m_{op}$ and the corresponding maximum relative running speed $V_{max}$. Figure 3i shows a three-dimensional surface constructed from the simulation results of the maximum relative running speed of the BHMbot across varying payload masses and body lengths. It indicates that a smaller body length correlates with a larger maximum relative running speed when the payload mass is 0 or $m_{op}$. However, a smaller body length will also lead to a smaller $m_{op}$ and a narrower range of payload mass for achieving relatively high running speeds. This conclusion is consistent with the experimental results. The scaling results and detailed analysis of the performance parameters are summarized in Supplementary Table 7 and expounded upon in Supplementary Note 8.

## Control strategy

Legged animals can readily change their moving direction by adjusting the actuation difference between the left and right legs. In this work, direction control is achieved by adjusting the two independent driving channels applied to the two electromagnetic actuators. Figure 4a–c show three fundamental movements of the BHMbot (prototype #2.2) under corresponding driving channels: straight running, clockwise turn, and anticlockwise turn (Supplementary Movie 5). The blue channel 1 is applied to the left actuator, controlling the left front leg; while the orange channel 2 is applied to the right actuator, controlling the right front leg. When channel 1 and 2 are both effective, two front legs are both actuated and the BHMbot runs forward. When channel 1 is effective and channel 2 is ineffective, the left front leg is actuated and the BHMbot turns clockwise with the other three legs serving as pivot points. Similarly, when channel 1 is ineffective and channel 2 is effective, the right front leg is actuated and the BHMbot turns anticlockwise.

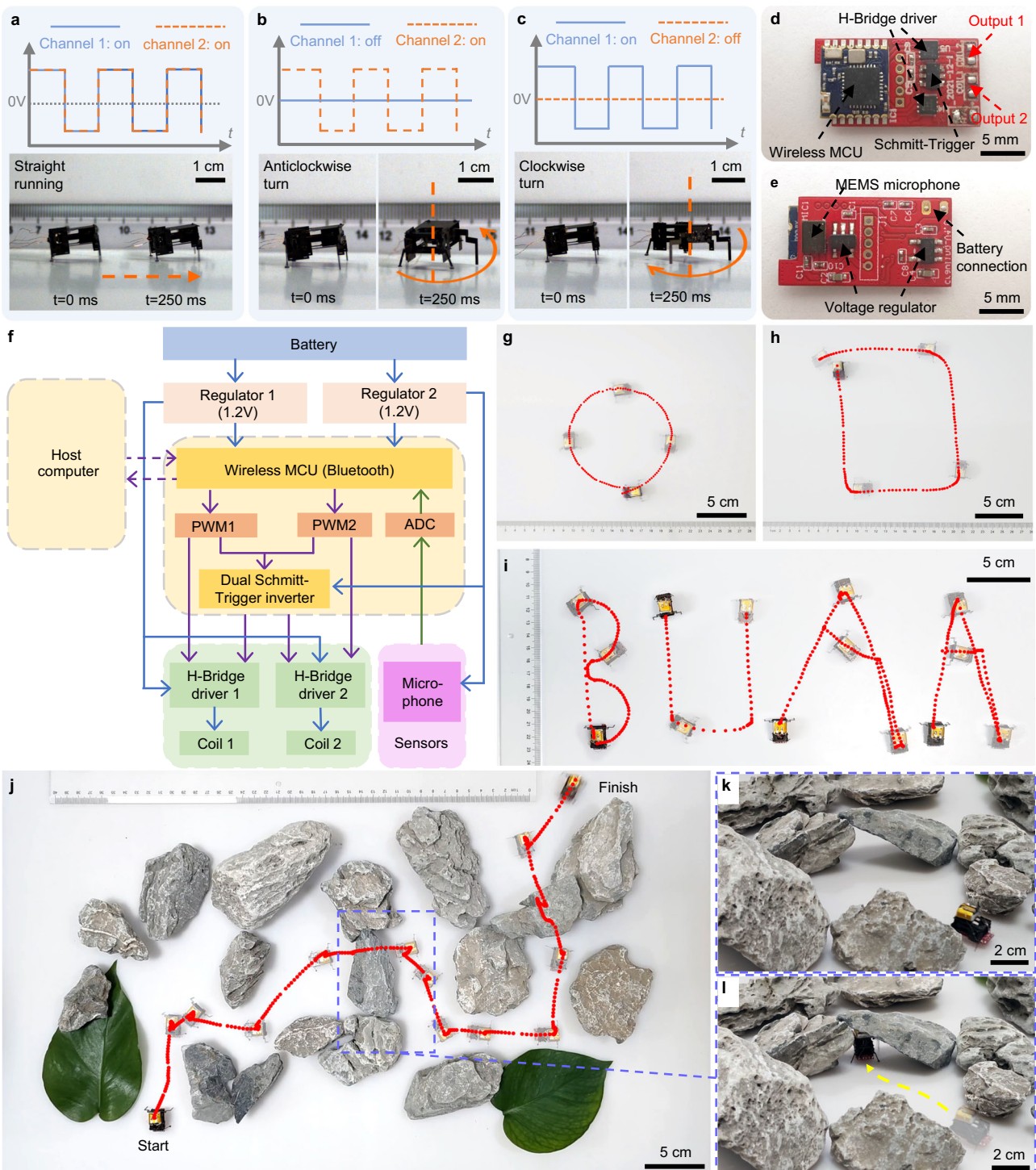

**Fig. 4 | Trajectories control of the BHMbot. a–c** Driving channels and corresponding optical photos from a high-speed camera showing that the BHMbot (prototype #2.2) achieves a straight movement under two in-phase driving channels, a clockwise turn, and a counterclockwise turn within a period of 250 ms, respectively. The blue line represents channel 1 and the orange line represents channel 2. **d**, **e** Top view and bottom view of the power and control circuit board. **f** Block diagram illustrating the design of the power and control circuit. Blue lines indicate driving signals. Purple lines indicate control signals. Green lines indicate sensor signals. **g** Optical photo of the BHMbot (prototype #5) running along a circle with a diameter of 10.5 cm in 4.5 s. **h** Optical photo of the BHMbot (prototype #5) running along a rectangle with a length of 11.5 cm and a width of 14 cm in 8.0 s. **i** Optical photo of the BHMbot (prototype #5) running along four capital letters "BUAA" (abbreviation of the full name of BeiHang University). The consumed times for B, U, A, and A are 8.0 s, 6.8 s, 16.2 s, and 13 s, respectively. **j** The scenario of the BHMbot (prototype #5) crossing a 40 cm-by-60 cm area scattered with barriers in 70 s by sending real-time commands. **k**, **l** Optical photos showing the BHMbot (prototype #5) passing through a narrow tunnel (2.5 cm × 4.0 cm). The yellow dotted line represents the trajectory of the BHMbot.

In addition to on-off switch control, trajectory control can also be achieved by adjusting the frequency and the amplitude of the two driving channels (Supplementary Fig. 5a, b). The relative linear running speed and turning centripetal acceleration of the BHMbot reach the maximums when the driving frequency is close to the resonant frequency of the BHMbot. Apart from the running speed and turning centripetal acceleration, test results also indicate that the turning radius of the BHMbot can be controlled by adjusting the frequency difference between two driving channels (Supplementary Fig. 5c−f). A larger frequency difference will result in a smaller turning radius. When the frequency of one channel drops to 0 and the other one is close to the resonant frequency, the BHMbot achieves 300° clockwise and 320° anticlockwise turns with radii of 1.0 cm and 0.7 cm in 0.4 s, respectively (Supplementary Movie 6). With a frequency difference of 50 Hz, the BHMbot achieves 90° right and left turns with radii of 8.1 cm and 7.9 cm in 0.8 s (Supplementary Movie 6). A continuous movement consisting of two linear paths and a 90° turn is shown in Supplementary Fig. 5g (Supplementary Movie 6).

To demonstrate wireless controlled untethered locomotion, a miniaturized power and control circuit is developed based on the frequency control strategy (600 mg, 2.0 cm by 1.0 cm), as shown in Fig. 4d, e (see the Methods Section). Since the electromagnetic actuators can operate under a relatively low alternating voltage (1.2 V), booster circuits are no longer necessary for the circuit design. Furthermore, the proposed control strategy requires only two independent channels, which also simplifies the circuit design. Figure 4f illustrates the schematic of the circuit, which consists of two voltage regulators, a Bluetooth micro control unit, a dual Schmitt trigger, and two H-bridge drivers. The circuit is powered by a lithium battery (50 mAh, 3.7 V, 800 mg) and can output two driving channels with variable frequencies under wireless commands from a computer or a smartphone.

Based on the circuit, an untethered BHMbot (prototype #5, 2-cm-long, 1700 mg) is fabricated to demonstrate wireless trajectory control capability. It is noted that the body length of prototype #5 without the circuit board is still 1.5 cm (same as prototype #2.2). The absolute speed, turning radius, and turning centripetal acceleration of the BHMbot are measured on a plastic board (Supplementary Fig. 6c−e). Basic movements such as left and right turns with an expected radius and straight running can be easily achieved by adjusting the frequency difference between two driving channels. Figure 4g, h demonstrate the BHMbot moves along a circular trajectory with a diameter of 10.5 cm (Supplementary Movie 7) and a rectangle trajectory with a length of 11.5 cm and a width of 14 cm (Supplementary Movie 7). Figure 4i shows that the BHMbot moves along more complicated paths under a set of pre-programmed control commands (Supplementary Movie 8): four capital letters BUAA (abbreviation of BeiHang University).

When the BHMbot needs to move along an irregular path to avoid barriers, it is more appropriate to send real-time commands to control its motion. To this end, a smartphone application is developed for remote control of the BHMbot. Figure 4j shows the BHMbot goes through a 40 cm-by-60 cm area scattered with stones and leaves under remote control from a smartphone (Supplementary Movie 9). During a total running period of 70 s, the BHMbot achieves 27 motion changes and moves along an irregular path with a total distance of 73.0 cm. The smallest tunnel on the path is 4.0 cm wide and 2.5 cm high. (Fig. 4k, l).

### Untethered locomotion performance evaluation

To evaluate the untethered locomotion performance of the BHMbot quantitatively, we select the relative linear running speed and relative turning centripetal acceleration as two performance indicators[17]. The relative turning centripetal acceleration $a_r$ characterizes the turning agility of the BHMbot and can be expressed as:

$$a_r = \frac{v^2}{R_t} \frac{1}{BL} \qquad (3)$$

where $v$ is the absolute moving speed; $R_t$ is the turning radius; and BL is the body length of the BHMbot. An untethered BHMbot (prototype #6, 1760 mg, 2.0-cm-long) with optimized parameters and a MEMS microphone is fabricated to test the untethered performance on four different surfaces, including a glass board with a friction coefficient $\mu = 0.161$, a wooden desktop with $\mu = 0.204$, a standard printing paper with $\mu = 0.320$ and a plastic board with $\mu = 0.372$. The measurements of friction coefficients of the four surfaces are shown in Supplementary Note 9, Supplementary Fig. 7, and Supplementary Table 8.

Due to manufacturing and assembly errors, the BHMbot tends to achieve straight running movements under a small frequency difference (30−40 Hz on four surfaces). The BHMbot achieves the fastest speed of 17.5 BL s$^{-1}$ on the paper surface (Fig. 5a and Supplementary Movie 10). During the turning tests, only one actuator is activated to achieve the minimum turning radius and maximum turning centripetal acceleration. The BHMbot achieves the maximum relative centripetal acceleration of 39.4 BL s$^{-2}$ in the clockwise turning and 65.4 BL s$^{-2}$ in the anticlockwise turning on the paper surface (Fig. 5b, c, and Supplementary Movie 11). The test results on the other three surfaces are shown in Fig. 5d, e (Supplementary Fig. 8a−c). The detailed data used to calculate $a_r$ is shown in Supplementary Table 9, including the turning radius, the absolute running speed, and the frequency of driving channels.

Figure 5f presents the comparison of the relative moving speed with respect to the body mass for several terrestrial mammals[24–26] (blue triangles), insects[2,3,27–32] (orange circles), reported untethered robots[8–10,16,17,22,23,33–45] (purple diamonds), and the BHMbot (red pentagram). Detailed data for comparison is provided in Supplementary Table 10. It is observed that mammals and insects demonstrate a higher relative moving speed at the same body mass level compared to the reported untethered robots. The BHMbot achieves a higher relative running speed compared with other untethered insect-scale microrobots, even at a larger mass level. Although the relative running speed of 17.5 BL s$^{-1}$ of the BHMbot is slower than the speed of the fastest cockroach[28] (50 BL s$^{-1}$), it surpasses the performance of a common cockroach (Nauphoeta cinerea[27], 13 BL s$^{-1}$) and several kinds of other insects such as Eremobates marathoni[31] (9.9 BL s$^{-1}$).

Figure 5g shows the comparison of relative centripetal acceleration with respect to body length for several terrestrial mammals[46–49] (blue triangles), insects[50–54] (orange circles), reported untethered robots[33,34,55–59] (purple diamonds), and the BHMbot (red pentagram). Detailed data for comparison is provided in Supplementary Table 11. For mammals, insects, and artificial robots, larger relative centripetal accelerations are typically achieved with smaller body sizes due to the decrease of inertia effect. The BHMbot presented in this work exhibits higher turning agility than the reported untethered microrobots at the same size scale. The relative centripetal acceleration at 65.4 BL s$^{-2}$ achieved by the BHMbot exceeds the turning performance of several reported untethered robots even when the body length is extended to 1 m. This turning agility also exceeds the performance of a cockroach (Blaberus discoidalis[53], 14.32 BL s$^{-2}$) and other insects at the same body size level such as honey bees[51] (5.27 BL s$^{-2}$).

Additionally, a plastic board with shallow puddles of water and a plastic round pipe are chosen as additional surfaces to further showcase the locomotion performance of the BHMbot. The BHMbot can run unhindered with a relative speed of 10 BL s$^{-1}$ on the plastic board with puddles of water (Supplementary Fig. 8d and Supplementary Movie 12). To test the locomotion performance of the BHMbot on a

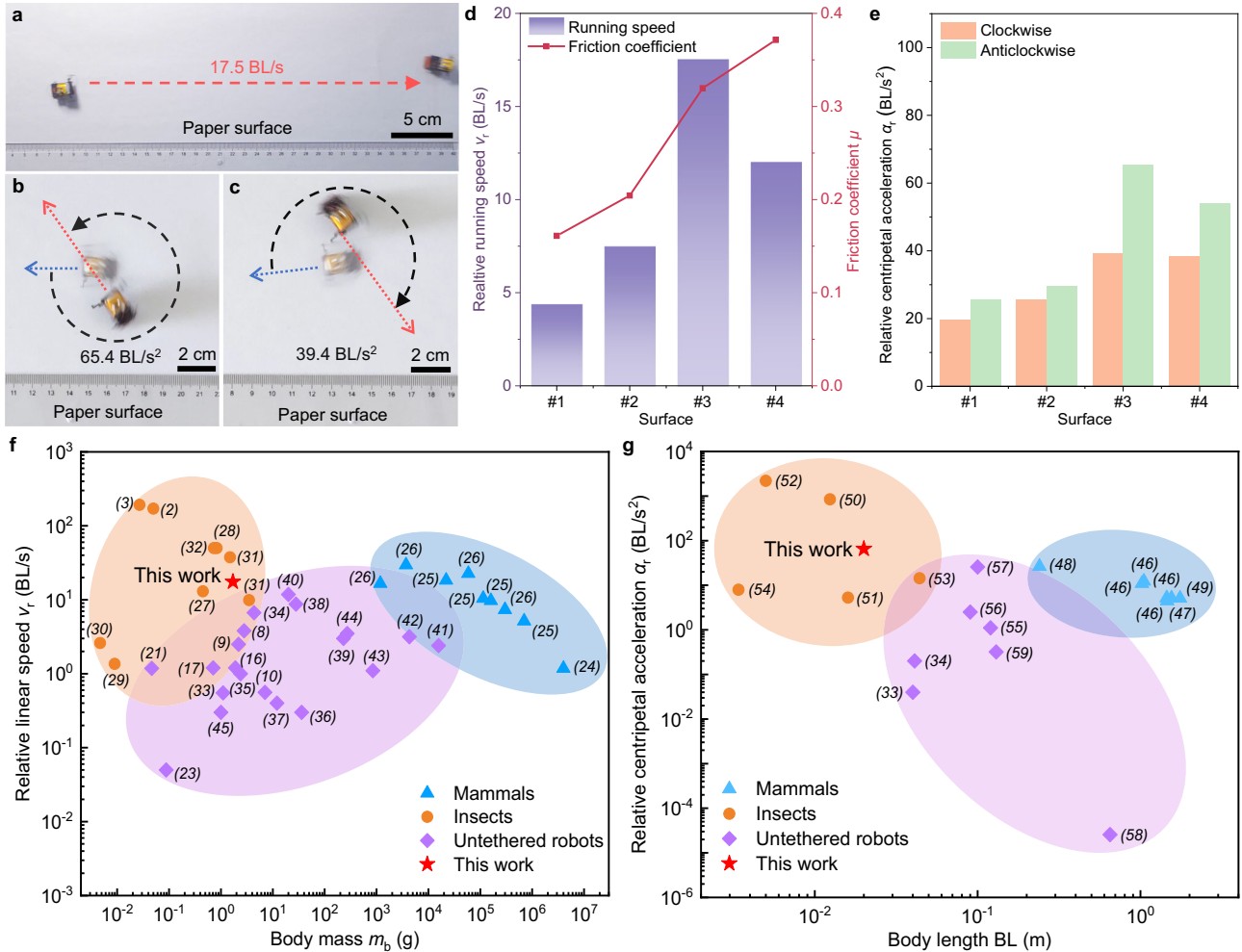

**Fig. 5 | Untethered locomotion performance of the BHMbot (prototype #6) and comparison with natural creatures and other untethered robots. a** Optical photos of the BHMbot running linearly on a paper surface with a maximum speed of 17.5 BL s$^{-1}$. The red dotted lines represent the locomotion displacements of the BHMbot in 1.0 s. **b**, **c** Optical photos of the BHMbot achieving anticlockwise and clockwise turns on the paper surface with a maximum relative centripetal acceleration of 65.4 and 39.4 BL s$^{-2}$ respectively. The gray dotted lines represent the angular displacements of the BHMbot in 0.5 s. **d** Maximum relative running speeds of the BHMbot on four different surfaces with different friction coefficients $\mu$, numbered from #1 to #4, corresponding to glass, wood, paper, and plastic surfaces. **e** Maximum centripetal accelerations of the BHMbot during clockwise (orange bars) and anticlockwise (green bars) turns on the four surfaces. **f** Comparison of the relative linear running speed versus the body mass with selected natural mammals (blue triangles), insects (orange circles), other untethered robots (purple diamonds) and the BHMbot (red pentagram). **g** Comparison of the relative centripetal acceleration versus the body length with selected mammals (blue triangles), insects (orange circles), other untethered robots (purple diamonds) and the BHMbot (red pentagram).

curved surface, the BHMbot is put in a round pipe with an inner diameter of 10 cm and the relative speed of the BHMbot is 2.5 BL s$^{-1}$ (Supplementary Fig. 8e and Supplementary Movie 12).

The high running and turning agility of the BHMbot can be attributed to the complementary combination of the bouncing length and high bouncing frequency. In this work, the cantilever in the electromagnetic actuator and the transmission of the BHMbot can be designed with relatively high stiffness, which leads to high resonant frequencies under different payloads (Supplementary Table 5).

## Cost of transport

In addition to kinematic indicators, energy efficiency is also a key indicator, and it is characterized by the Cost of Transport[60] (COT). The value of COT is determined by the power consumption, mass, and relative speed of the BHMbot:

$$\text{COT} = \frac{P}{mg\nu} \quad (4)$$

In this work, we select two COTs (COT$_T$ and COT$_M$) to estimate the efficiency of the whole microrobot and the actuation mechanism for running gait respectively (see Supplementary Note 10). Supplementary Fig. 9 shows the power flow of an untethered BHMbot (prototype #6). The calculation of COT$_T$ concentrates on the power flow from the battery to the microrobot, and the power $P$ is equal to the output power of the battery $P_b$. For prototype #6, the measured $P_b$ is 1.77 W and the value of COT$_T$ is 303.7. When the battery capacity is 50 mhA, the measured maximum duration of prototype #6 can reach 3 min (Supplementary Movie 13).

To evaluate the energy efficiency of the actuation mechanism, we also calculate COT$_M$ which only takes into account the power flow from the actuators to the microrobot, and $P$ is equal to the output power of two electromagnetic actuators $P_a$. The COT$_M$ can be given as:

$$\text{COT}_M = \frac{P_a}{mg\nu} = \frac{4f_a \int_{-s_0}^{s_0} F_{\text{EM}} ds}{mg\nu} \quad (5)$$

where $f_a$ is the vibration frequency of the actuators (220 Hz for prototype #6); $F_{EM}$ is the electromagnetic force of the actuators; $s$ is the displacement of the magnet relative to the coil and $s_0$ is the maximum displacement of the magnet during the vibration process. The measured $P_a$ is 5.62 mw, and the value of $COT_M$ is 9.3, which indicates the high efficiency of the actuation mechanism.

In particular, we perform a comprehensive comparison of the BHMbot with six other insect-scale legged microrobots presented in recent years. Table 1 shows the performance indicators of this work and other six microrobots. The $COT_T$ of the BHMbot is higher than the piezoelectric-driven microrobots but lower than the soft microrobots. The energy efficiency of the BHMbot is primarily influenced by the large current flowing through the circuit board and the low actuation efficiency of the electromagnetic actuators[19]. When focusing solely on the efficiency of the actuation mechanism, the $COT_M$ of the BHMbot drops to 9.3 and it is close to the minimum value among reported insect-scale microrobots. Considering that the actuation mechanism is also applicable to other linear actuators, the relatively low $COT_M$ indicates the potential of the presented actuation mechanism as a promising option for other insect-scale legged microrobots.

### Application scenarios demonstration

Aided by the mini size and high mobility, the BHMbot can go through narrow spaces and reach specified locations to execute tasks, such as search and rescue missions and inspection of inner structures inside turbofan engines. In practical scenarios where the task location is relatively far away from the starting point, the BHMbot can be transported by a drone to realize a quick arrival at a nearby area of the destination.

To demonstrate potential applications of the BHMbot in rescue missions, we set up a scenario shown in Fig. 6a. A simulated collapsed structure built of wooden blocks is placed on the right side of a plastic board. A Bluetooth speaker is buried in the house and sends out SOS signals. The BHMbot (prototype #6) is controlled remotely to navigate obstacles on the plastic board (such as stones and puddles) and halt in proximity to the structure (red trajectory). A commercial MEMS microphone (60 mg) is integrated into the onboard circuit to collect sound signals. The waveform of the collected sound signals by the MEMS microphone can be visualized using a smartphone application. Upon acquiring the SOS signals, the BHMbot returns to the starting point along another route (blue trajectory). The sound data is transmitted to a computer and converted to real sound. The original SOS signals sent by the Bluetooth speaker and the recovered signals from the computer are shown in Fig. 6b. The detailed process is shown in Supplementary Movie 14.

Considering the high mobility of the BHMbot, it also provides a promising platform for conducting structural inspection tasks within aero engines. Figure 6c shows the BHMbot goes through a narrow passage between two stator blades of a turbofan engine and then returns to the starting point (Supplementary Movie 15). Figure 6d shows the BHMbot runs quickly with a maximum speed of 4.5 BL s⁻¹

within the tail cone of a turbojet engine (Supplementary Movie 15). Benefiting from the development of micro cameras at millimeter scale[61], it is promising for the BHMbot to integrate with a micro camera to capture internal images of aero engines in the future.

Due to limited battery life and inability to climb, the BHMbot faces challenges in traveling long distances or reaching high geographical positions. In such scenarios, a small quadrotor can transport the BHMbot to a location near the target area and retrieve it once the mission is accomplished. As shown in Fig. 6e, a quadrotor equipped with a nacelle is utilized to transport the BHMbot from the ground to the desktop. Figure 6f shows that the BHMbot exits the nacelle, moves around a stone, and then returns to the nacelle for transportation back to the starting point by the quadrotor (Supplementary Movie 16). Through collaborating with quadrotors, the BHMbot exhibits promising potential for search and rescue applications in the near future.

## Methods

### Fabrication of electromagnetic actuator

The linear electromagnetic actuator is composed of a permanent magnet, a hollow coil and a cantilever that serves as an elastic restoring component. Four cylindrical permanent magnets with different sizes (Supplementary Table 4) are utilized in the design of the electromagnetic actuators for prototypes #2.1–2.4. All the magnets are made of NdFeB (N52). There are also four customized cylindrical hollow coils with different sizes (Supplementary Table 4) that are used to match the sizes of the magnets. All the hollow coils are made of copper wires with a diameter of 0.08 mm. To reduce the interaction between two magnets in two electromagnetic actuators, four baffles are added around the hollow coil, which are cut from a 300-μm-thick stainless-steel sheet (Feintool company). The cantilever is cut from sheets laminated with three orthogonal 50-μm-thick carbon fiber layers (Toray company). A hole with the same diameter as the magnet is cut out at the free end of the cantilever for fixing the magnet.

### Support frames

The support frames serve to secure other components, including the actuators, the transmissions and the onboard circuit. As illustrated in Fig. 7a, the majority of supporting frames are cut from 150-μm-thick carbon fiber sheets using a laser cutting machine (Han's Laser UV3C, 3 W in power, and 355 nm in wavelength, China). These sheets are composed of three orthogonal 50-μm-thick carbon fiber layers (Toray company). The rear legs are made of the same material. To mitigate the risk of short circuits, the support frame for the circuit board is cut from an insulated 100-μm-thick plastic film (Nalifilm company).

### Fabrication of transmission mechanism

The transmission mechanisms of the BHMbot are fabricated by applying the SCM (Smart Composite Microstructure Fabrication) method[62] and laser cutting techniques, which are integrated with the front legs to reduce assembly errors. Figure 7b shows that the transmission mechanism is cut from a 245-μm-thick composite laminate

**Table 1 | Comparison of this work and six other insect-scale untethered microrobots with mass less than 5 g and body length less than 5 cm**

| Microrobot description | Length (mm) | Total mass (g) | Speed (BL s⁻¹) | Turning centripetal acceleration (BL s⁻²) | Cost of Transport (COT_T) | Cost of Transport (COT_M) |
|---|---|---|---|---|---|---|
| This work | 20 | 1.76 | 17.5 | 39.4/65.4 | 303.7 | 9.3 |
| HAMR-F[8,14] | 45 | 2.8 | 3.8 | 16.4 (tethered) | 83.9 | 13.0 |
| DEAnsect[33] | 40 | 1 | 0.3 | 0.04 | 1670 | 5.3 |
| S²worm[34] | 41 | 4.34 | 6.7 | 0.20 | 52.4 | – |
| PVDF robot[17] | 24 | 1.9 | 1.2 | 0.09 | 887 | – |
| RoBeetle[23] | 15 | 0.088 | 0.05 | – (uncontrollable) | – | – |
| SEMR UR1[16] | 20 | 2.2 | 2.1 | – | – | – |

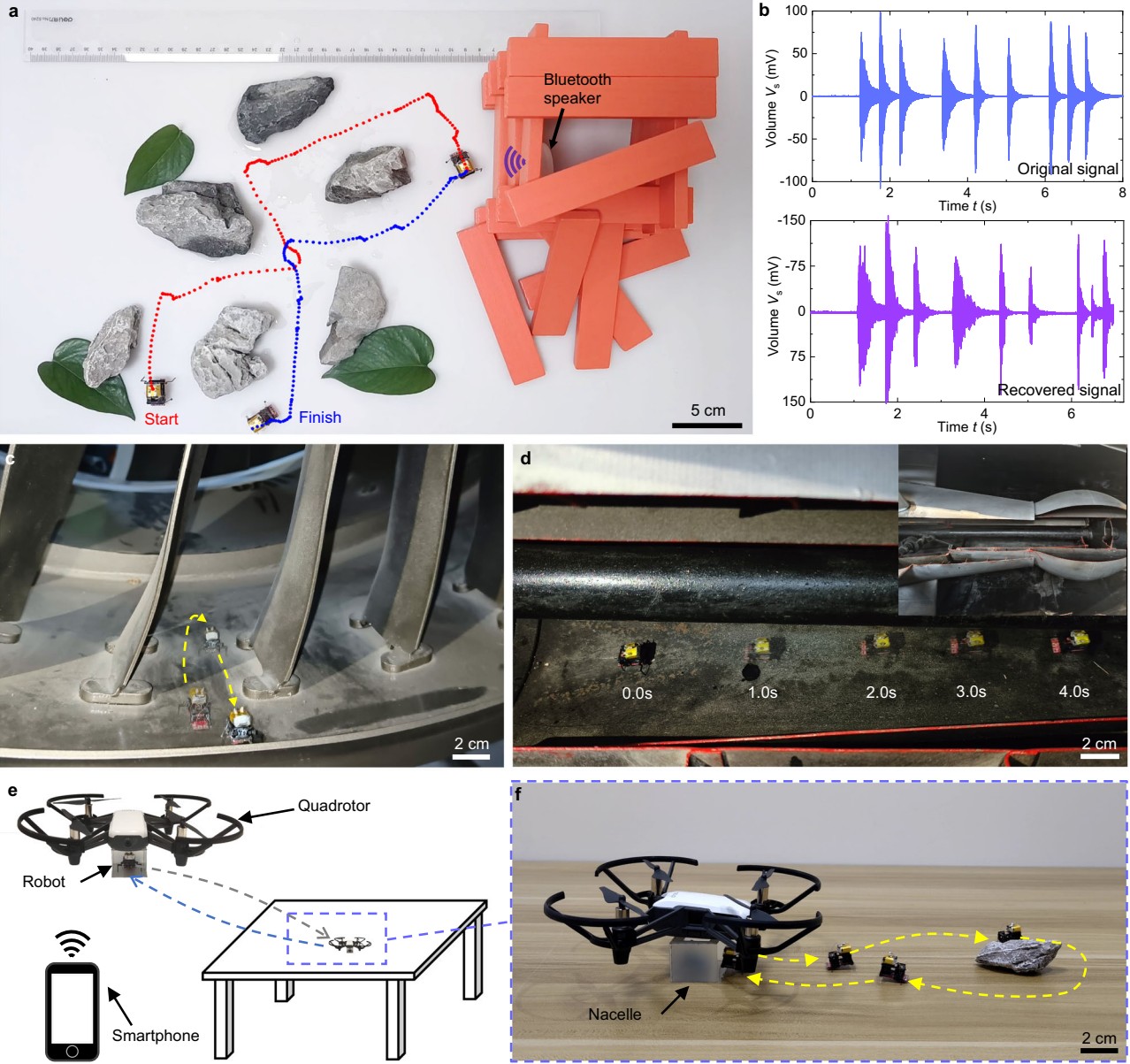

**Fig. 6 | Application scenarios demonstration of the BHMbot (prototype #6).**
**a** Demonstration of the BHMbot running through a complex area scattered with obstacles to a collapsed house to collect SOS signals (red trajectory) and then returning to the starting point (blue trajectory). **b** Waveforms of the original SOS signals from the Bluetooth speaker (blue line) and the recovered SOS signals from the data collected by the BHMbot (purple line). **c** Optical photo showing the BHMbot running through the passage between two stator blades of a civil turbofan engine. The yellow dotted line represents the moving trajectory of the BHMbot. **d** Optical photo showing the BHMbot running on the inner surface of the tail cone of a turbofan engine. **e** Application scene where the BHMbot is transported to a desktop by a quadrotor to execute tasks and then transported to the start point. **f** Optical photo showing that the BHMbot moves out of the nacelle of the quadrotor, moves around a stone, and then returns to the nacelle. The yellow dotted line represents the moving trajectory of the BHMbot.

consisting of two 100-µm-thick carbon fiber layers (top and bottom), two 12.5-µm-thick sheet adhesive layers (DuPont™ Pyralux® LF1500) and a 20-µm-thick polyimide film layer (DuPont™ Kapton® CR). The sheet adhesive layers are utilized to bond the carbon fiber layers and the polyimide film layer under high temperature (300 °C) and pressure conditions.

### Assembly process of the BHMbot
The 2D structures are manually assembled and folded to form a 3D prototype, as illustrated in Fig. 7c. The support frames are glued together with the assistance of mounting grooves and flanges to guarantee precise assembly. The transmission mechanism is folded into a 3D structure and then affixed to the magnet and cantilever via a locating hole. Subsequently, the three components are adhered to the rear and front support frames using mounting grooves and flanges. The detailed dimensions of the six 15-mm prototypes in this paper are provided in Supplementary Table 12.

### Power and control electronics
A miniaturized power and control circuit board weighing 600 mg is developed using commercial components to realize the untethered and controlled locomotion of the BHMbot. On the top side of the board, a wireless microcontroller unit (MCU), two H-Bridge drivers, a Schmitt-Trigger inverter, and four output pads are soldered onto the PCB board (Fig. 4d). The four output pads can provide two independent channels with variable frequencies. On the back side, there are two voltage regulators, a MEMS microphone, and two input pads for battery connection (Fig. 4e). The surface of the PCB board is 143 mm².

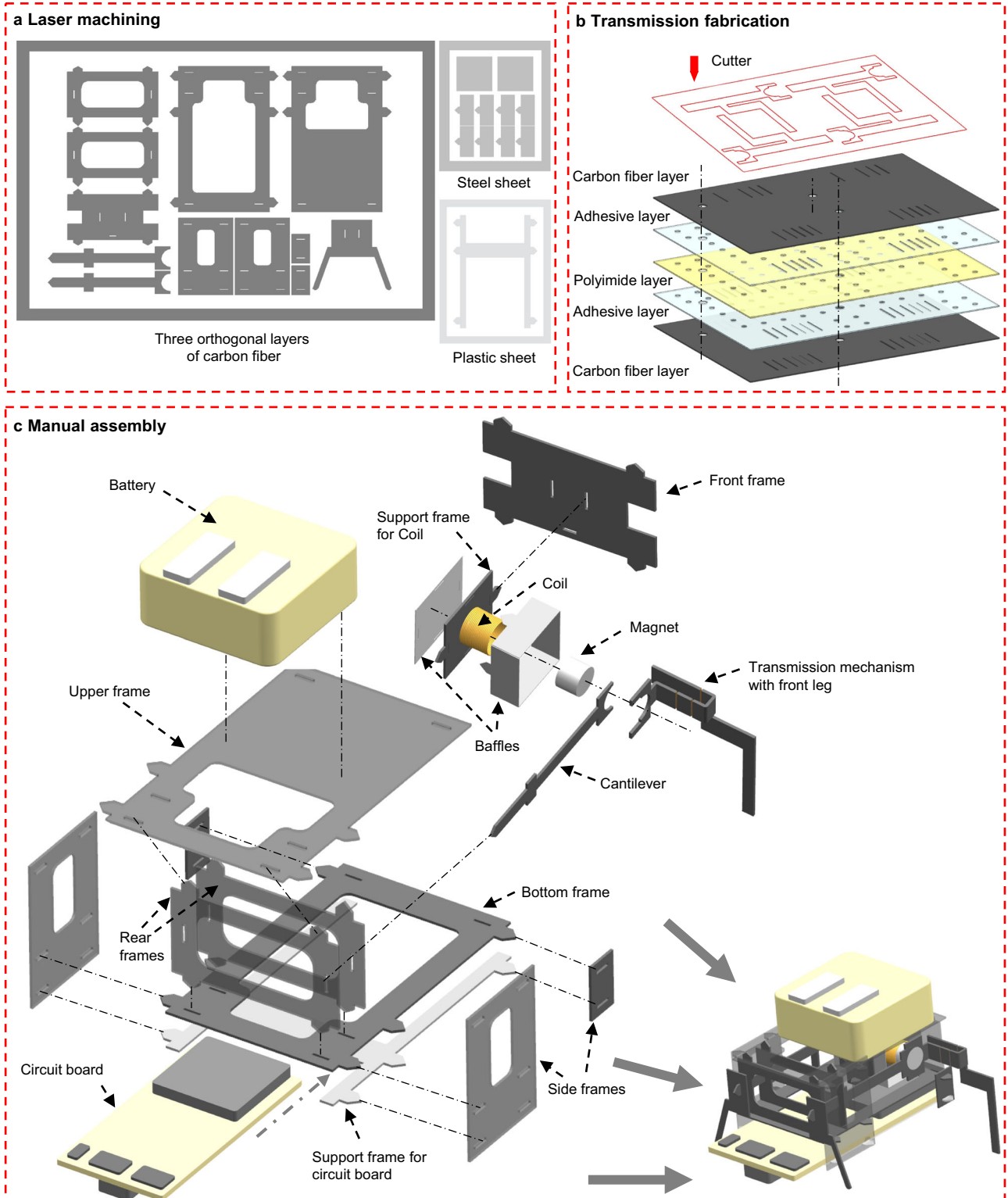

**Fig. 7 | Laser cutting and assembly process of the BHMbot. a** Fabrication process of the planar structures via laser cutting. A thin sheet laminated by three orthogonal layers of carbon fiber is selected as the material for the supporting frames, cantilever, and rear legs because of their high strength. A sheet of stainless steel is used to fabricate the baffles around the coils because of its magnetic properties. The supporting frames of the electronics board are made up of a plastic sheet because of its insulating properties. **b** SCM fabrication process of the transmission mechanism. **c** Manual assembly process of the untethered prototype BHMbot with a length of 2 cm and a mass of 1760 mg.

The mass and quantity of the electrical components used in the circuit are shown in Supplementary Table 13.

The schematic design of the circuit is depicted in Fig. 4f. A lithium battery supplies two voltage regulators with a DC voltage of 3.7 V. Regulator 1 reduces the input voltage to 1.2 V and outputs it to two H-Bridge drivers. Regulator 2 lowers the input voltage to 3.3 V to power the micro controller and the MEMS microphone. The wireless micro-controller unit, integrated with a Bluetooth communication module, receives commands from a Bluetooth host (such as a computer or a smartphone) and generates two PWM signals (PWM1 and PWM2). It also performs the A/D conversion and transmission of the sound data received by the microphone. The PWM1 and PWM2 signals are sent to the Schmitt-Trigger inverter, which outputs two inverting PWM signals (PWM3 and PWM4). These four PWM signals are directed to two H-Bridge drivers to acquire the essential driving channels for two electromagnetic actuators. Supplementary Fig. 6a shows the detailed schematic diagram illustrating the generation process of the two driving channels. The waveforms of four PWM signals and two driving channels are shown in Supplementary Fig. 6b. The frequencies of the two driving channels are controlled by PWM1 and PWM2 respectively and the amplitudes remain unchanged (1.2 V). With the Bluetooth communication module, we can control the locomotion of the BHMbot via two schemes. One scheme is to automatically send a set of commands at programmed intervals to guide the BHMbot along desired trajectories. The other scheme is to send real-time commands manually to continually adjust the trajectory of the BHMbot based on the image observed by the operator.

## Data availability

All data needed to evaluate the conclusions in the paper are presented in the paper and the Supplementary Information. Additional data related to this paper is available upon any request.

## Code availability

The code used in this paper is available upon any request.

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

## Acknowledgements

This work is supported by the Beijing Natural Science Foundation (Grant No. 3232010, Z.L.), the National Natural Science Foundation of China (Grant No. 12002017, Z.L.), the BeiHang Outstanding Young Scholars Project (NO. YWF-23-L-1201, Z.L.), and the 111 Project (Grant No. B08009, X.Y.). Any opinions, findings, conclusions, or recommendations expressed in this material are those of the authors and do not necessarily reflect the views of the National Natural Science Foundation of China.

## Author contributions

Z.L. and X.Y. proposed and designed the research. W.Z. and Z.L. designed and built the BHMbot. W.Z., Z.L., Y.Z., X.L. and L.W. conducted the experimental work. X.W. and S.H. designed the power and control electronics. J.L. and M.Q. contributed to the modeling and data analysis. W.Z. and Z.L. drafted the manuscript. X.Y. revised the manuscript. All authors provided feedback.

## Competing interests
The authors declare no competing interests.
