## [Peer Review File · Nature Communications]

Reviewers' comments:

Reviewer #1 (Remarks to the Author):

The paper presents a new biologically-inspired terrestrial locomoting microrobot, the BHMbot. This robot can function at relatively high speeds without tethers for power. The manuscript is generally well written in the sense that it is understandable and sufficiently solid from a scientific point of view. However, the manuscript requires improvements from the grammar and style viewpoints.

The authors argue that a key element of their design was the development of miniaturized active leg actuation (ALA) methods, which can take advantage of system-ground interaction phenomena (bouncing moment). Furthermore, the authors claim that the ALA approach enabled them not only to increase operational speeds but also payload capacities. One of the most impressive elements of the presented research is the control system and integration of all the required electronics on board. Specifically, if I understood the text correctly, no specialized microchips were developed for control and power. All the electronics parts are commercially available and, therefore, the research community can use and replicate the capabilities presented in the paper.

In my assessment, the BHMbot, was demonstrated, through a series of experiments, capable of speeds and agility comparable to those observed in natural insects. However, these capabilities were tested in relatively structured laboratory-like conditions. Despite these limitations, considering its actuation, speed of locomotion, and control, the BHMbot represents significant research progress in the field of mobile microrobotics. Mobile microrobotics is still a nascent field and even though the BHMbot is far from capable of emulating and replicating the capabilities of real insects, its development opens new avenues for exploration and research. In summary, I commend the authors for the presented research.

Last, I see only one major issue to be addressed before publication. The authors explicitly state that the electromagnetic actuators used to drive the BHMbot were designed and fabricated using commercially available parts. However, given their critical importance in their design, they should discuss the issue of actuation in a separate subsection. I would like to see descriptions of the fabrication process, functionality, efficiency, etc.

Reviewer #2 (Remarks to the Author):

The authors report a 2 cm long legged robot that moves at up to 17 BL/s in an untethered configuration. The robot trajectory can be well controlled. The quality of the English writing is very good. The overall robot performance is impressive in terms of speed.

While the robot has interesting overall speed, I have a hard time taking the article seriously due to the implausible and repeated claims of bioinspiration. How is this device more than superficially similar to a grasshopper? There is no evidence the design is based on the leg of a grasshopper. Grasshopper legs are optimized to jump. This robot does not jump. At best it bounces

The abstract (400 words long?) is not credible in my opinion: so much hype. Not a word on power consumption or efficiency.

The paper should take a more scientific and rigorous approach to discussing how the robot was designed. Why use EM actuation (rather eg than a piezo like for RoboBee) ? What force is needed to reach different gaits? What stroke? Heating effects?

What limits the speed? Why can't it move faster? What would allow faster motion? Is there a scaling argument?

Why does it move fast? How can you prove the bouncing helps? How were the linkages designed. Does the friction of the feet matter?

What performance on a slope?

How much power is needed? I can't find a number in mW. How was COT computed?

It should be made clearer how this is different from the authors earlier work (ref 11), images look similar to this work.

Not enough info on how sizes were determined.

I recommend submitting to "IOP Bioinspiration & Biomimetics"

Response to reviewers' comments

The previous manuscript entitled **“BHMbot: An Ultrafast Untethered Robotic Insect with Bioinspired Running Gaits Based on Active Leg Actuation”** (NCOMMS-23-31631) has been peer-reviewed by two reviewers during the last submission. The authors thank the reviewers for the effort in reviewing the manuscript and the editors for giving the opportunity to resubmit the manuscript to Nature Communications. We have considered all the comments carefully and addressed all of them in the new version of the manuscript entitled **“A Wireless Controlled Robotic Insect with Ultrafast Untethered Running Speed Based on Bioinspired Running Gait”**. Besides, we also provide a point-by-point response letter with substantial revisions to address the issues presented by the reviewers.

The authors sincerely thank all the reviewers for their valuable suggestions. We hope the revised manuscript can meet the publication criteria of Nature Communications. We appreciate your time and look forward to your response again.

Reviewer #1

The reviewer's comments: The paper presents a new biologically-inspired terrestrial locomoting microrobot, the BHMbot. This robot can function at relatively high speeds without tethers for power. The manuscript is generally well written in the sense that it is understandable and sufficiently solid from a scientific point of view. However, the manuscript requires improvements from the grammar and style viewpoints.

The authors' answer: We are very grateful for your insightful and detailed comments and these comments are of great importance in both revising this manuscript and guiding our research in the future. Based on your comments, we have revised the manuscript accordingly. The grammar errors have been checked and the overall language has been polished.

Suggestion #1: The authors argue that a key element of their design was the development of miniaturized active leg actuation (ALA) methods, which can take advantage of system-ground interaction phenomena (bouncing moment). Furthermore, the authors claim that the ALA approach enabled them not only to increase operational speeds but also payload capacities. One of the most impressive elements of the presented research is the control system and integration of all the required electronics on board. Specifically, if I understood the text correctly, no specialized microchips were developed for control and power. All the electronics parts are commercially available and, therefore, the research community can use and replicate the capabilities presented in the paper.

Responses #1: This comment is very insightful. The overall function of the electric circuit is converting the DC voltage of the battery to AC voltages required by the electromagnetic actuators under the control commands. The development of the circuit does not involve the design of microchips and the basic electronic components such as Bluetooth chips, resistors, capacitors and inductors that are commercially available, while the realization of the function of the electric circuit requires specific circuit design and programming. The circuit design can provide a reference for electromagnetically

actuated microrobots and the authors will be pleased if our work can contribute to the research community.

Suggestion #2: In my assessment, the BHMbot, was demonstrated, through a series of experiments, capable of speeds and agility comparable to those observed in natural insects. However, these capabilities were tested in relatively structured laboratory-like conditions. Despite these limitations, considering its actuation, speed of locomotion, and control, the BHMbot represents significant research progress in the field of mobile microrobotics. Mobile microrobotics is still a nascent field and even though the BHMbot is far from capable of emulating and replicating the capabilities of real insects, its development opens new avenues for exploration and research. In summary, I commend the authors for the presented research.

Responses #2: The authors sincerely thank the reviewer for the commendation. There is still a long way to go in achieving comparable capabilities of real insects for microrobots, and that is also a motivation for roboticists.

Suggestion #3: Last, I see only one major issue to be addressed before publication. The authors explicitly state that the electromagnetic actuators used to drive the BHMbot were designed and fabricated using commercially available parts. However, given their critical importance in their design, they should discuss the issue of actuation in a separate subsection. I would like to see descriptions of the fabrication process, functionality, efficiency, etc.

Responses #3: Thanks for your suggestion. Considering the importance of the electromagnetic actuator on the locomotion performance of the BHMbot, it is indeed necessary to discuss related issues of the actuator in detail, including the fabrication, mathematical modeling, parameter optimization, and energy efficiency.

(1) Fabrication. The fabrication of the actuator is mainly based on commercial products, including permanent magnets made of NdFeB (N52) and customized hollow coils made of copper wire. We have added the fabrication details in the **Method Section**

“Fabrication of the BHMbot”.

(2) Modeling and parameter optimization. To optimize the performance of the actuator, an electromagnetic force model is established to investigate the key parameters that determine the performance of the actuator, including the relative distance between the hollow coil and the magnet z , the geometric dimensions of the hollow coil and the magnet (r , l , R , and L). The optimization of the geometric dimensions (z , r , l , R , and L) is essentially a problem of solving the extremum of a multivariate function. We utilize the numerical method to solve the optimization problem and obtain the optimal values of the geometric parameters. The objective function for optimization is selected as the force efficiency (the ratio of the maximum electromagnetic force to the weight of the actuator). After optimization, the force efficiency of the actuator rises to 21.8, which demonstrates an enhancement of 20.8% compared to the value before optimization (16.6).

Considering the word limit of the manuscript, the discussions about the mathematical modeling and parameter optimization of the actuator are added in the **Supplementary Information Note S1 and S2**.

(3) Efficiency. Due to the induced heating effect caused by the high working AC current, the efficiency of the electromagnetic actuator is relatively low (1.7%), and such a result is close to the reported energy efficiency of the existing electromagnetic actuators at the same size scale¹. The calculation process of the efficiency is discussed in **Supplementary Information Note S10 “COT calculation for the untethered BHMbot”**.

Reviewer #2

The reviewer's comments: The authors report a 2 cm long legged robot that moves at up to 17 BL/s in an untethered configuration. The robot trajectory can be well controlled. The quality of the English writing is very good. The overall robot performance is impressive in terms of speed.

The authors' answer: We are very grateful for your insightful and detailed comments, and these comments are very helpful in both revising this manuscript and guiding our research in the future. We sincerely apologize to the reviewer for the unclarity in the last submission which fails to expound the scientific and technical advancement of the BHMbot. According to your suggestions, we have adjusted the structure of the manuscript and addressed all the comments. Before a detailed point-by-point response, we would like to provide a brief answer aiming at your three main concerns:

Major concern #1: The bioinspired design of the leg actuation.

Response: The comment regarding the bioinspired design of the BHMbot is very professional and provides a good guiding direction for the authors to reconsider the moving mechanism of the BHMbot. As the reviewer pointed out, the forward locomotion of the BHMbot through periodical bouncing movements is different from the jumping movements of grasshoppers. After carefully analyzing the bouncing movements of the BHMbot, we think the forward locomotion of the BHMbot through periodical bouncing movements is similar to the running postures of several mammals (such as cheetahs). We have rewritten the **Section "Design and moving mechanism"**, and Fig. 1 is also redrawn based on the **Section "Design and moving mechanism"**.

Major concern #2: The power consumption and efficiency (or cost of transport) of the robot.

Response: The authors apologize for the unclarity of power consumption and efficiency discussion in the last submission since the last version of the manuscript is transferred from the submission to Nature and some contents are given in

Supplementary Information due to the word limit. In the last submission, the power consumption and cost of transport (COT) of the untethered robot are discussed in the **Supplementary Information Note S5 “COT calculation for the untethered BHMbot”**. To highlight its importance, we have transferred this part to **an independent section “Cost of transport”** in the new manuscript. Furthermore, the related data about the power consumption and COT has been stated in the **Abstract**. We also add the discussion about the power consumption of the tethered prototype in the **Section “Tethered locomotion tests and parameters optimization”**.

Besides the COT analysis, other concerns including the slope performance and the influence of the friction of the feet are also mentioned in the last submission. We have also made revisions accordingly to make these contents clearer in the new manuscript.

Major concern #3: Lack of a more scientific and rigorous approach to discussing how the robot was designed.

Response: Thanks for your insightful comment regarding the scientific and rigorous analysis of the BHMbot. To address this issue, we provide a more rigorous and comprehensive discussion about the design of our robot, including the selection of the actuation, the design of the transmission, the influencing factors of the running speed, the method for the enhancement of the running speed, the inner mechanism of achieving the high-speed locomotion after carrying payloads, the scaling effects and size determination.

Point-by-point response to the reviewer’s suggestions

Suggestion #1: While the robot has interesting overall speed, I have a hard time taking the article seriously due to the implausible and repeated claims of bioinspiration. How is this device more than superficially similar to a grasshopper? There is no evidence the design is based on the leg of a grasshopper. Grasshopper legs are optimized to jump. This robot does not jump. At best it bounces.

Responses #1: As mentioned in the response to the major concern #1, the forward

locomotion of the BHMbot through periodical bouncing movements is different from the jumping movements of grasshoppers. Based on the high-speed images of the BHMbot during forward locomotion (Supplementary Movie S2), it is observed that the bouncing movements of the BHMbot (Fig. R1) is similar to the running postures of several mammals (such as cheetahs). We have rewritten the Section “Design and moving mechanism”, and Fig. 1 is also redrawn based on the Section “Design and moving mechanism”.

Fig. R1. The running locomotion of the BHMbot and a cheetah. **a**, Series of high-speed images showing the running gait of the BHMbot, including initial, squatting, bouncing, aerial, and landing phases. **b**, Illustration depicting the running gait of a cheetah².

Suggestion #2: The abstract (400 words long?) is not credible in my opinion: so much hype. Not a word on power consumption or efficiency.

Responses #2: As suggested, we have rewritten the abstract and added the statements of the power consumption and *COT*. The performance data in the abstract is supported by the supplementary videos.

The revised abstract is as follows.

“Running speed degradation of insect-scale (less than 5 cm) legged microrobots after carrying payloads (such as power and control units) has become a bottleneck for microrobots to achieve high untethered locomotion performance. To this end, we present a 2-cm legged microrobot (BHMbot, BeiHang Microrobot) with ultrafast untethered running speed based on a bioinspired running gait. The vibrating front legs of the BHMbot impact the ground periodically to provide the momentum for the microrobot to bounce off the ground, during which the BHMbot goes through an

apparent aerial phase with all legs in the air and the complementary combination of bouncing length and bouncing frequency contributes to its outstanding locomotion performance even with payloads several times its body mass. The untethered BHMbot (20-mm-long, 1760 mg), which is integrated with power and control units, can achieve a running speed of 17.5 BL/s (13 BL/s for a cockroach, *Nauphoeta cinerea*) and turning centripetal acceleration of 65.4 BL/s² (14 BL/s² for a cockroach, *Blaberus discoidalis*) at a cost of transportation of 303 and power consumption of 1.77 W. Such speed is the fastest speed ever reported among insect-scale legged microrobots to the best of our knowledge. By controlling the two front legs independently, the BHMbot demonstrates various locomotion trajectories including circles, rectangles, letters and irregular paths across obstacles through a wireless control module. Such advancements enable the BHMbot to carry out application attempts including SOS signal detection, locomotion in a turbofan engine and transportation via a quadrotor.”

Suggestion #3: The paper should take a more scientific and rigorous approach to discussing how the robot was designed. Why use EM actuation (rather eg than a piezo like for RoboBee)? What force is needed to reach different gaits? What stroke? Heating effects?

Responses #3.1 for actuation concern: Considering that the mass of the power and control circuit has a significant influence on the untethered locomotion performance of the microrobot^{3,4}, we select the electromagnetic actuator to drive the BHMbot because of its relatively low operating voltage at several volts (< 2 V). The booster circuit is no longer needed for the electromagnetic actuator as compared with piezoelectric actuators, which is beneficial to the lightweight design of the power and control circuit. From the perspective of mechanism design, the presented actuation mechanism also applies to other linear actuators, such as piezoelectric actuators used by the Harvard HAMR⁵, which means it can provide a practical solution for other insect-scale legged microrobots for enhanced untethered performance. We have discussed the issue in the **Section “Design and moving mechanism”**.

Responses #3.2 for gait concern: As mentioned in response #1, the locomotion of the BHMbot through periodical bouncing movements (Fig. R1) is similar to the running postures of several mammals (such as cheetahs), instead of the jumping postures of grasshoppers. To achieve the desired bouncing movements, the front legs of the BHMbot are intentionally designed to be longer than the rear legs to form an upward tilt angle θ_0 of the body. Therefore, the backward swing motion of the front legs will encounter resistance from the ground. In response, the front leg pushes off the ground, generating an obliquely upward reaction force F' , as shown in Fig. R2a. Consequently, the BHMbot bounces off the ground under the action of F' and then lands on the ground (Fig. R2b). Fig. R2c shows the measurements of the vertical position of the rear and front feet of the BHMbot during one bouncing cycle. We have discussed the issue in the Section “Design and moving mechanism” in the new manuscript.

Fig. R2. Moving mechanism of the BHMbot. **a**, Diagrams illustrating the generation of bouncing momentum for the BHMbot during an actuation cycle of the front leg. **b**, Diagram showing the bouncing movement of the BHMbot. **c**, Measurements of the vertical position of the rear and front feet of the BHMbot during one bouncing cycle. When all the rear and front feet are off the ground, the BHMbot is in the aerial phase (the blue area).

Responses #3.3 for heating effects: Due to a large working current (0.1~0.2 A), the induced heating effect of the hollow coil of the electromagnetic actuator is inevitable, which leads to a relatively low actuation efficiency. According to the reference¹, the energy efficiency of the linear electromagnetic actuator is less than 1% at the milligram

scale (an estimation based on the experimental data in the reference). We also obtain the efficiency of the electromagnetic actuator used in our robot based on the input and output power of the actuator (1.7%). This issue is discussed in the Supplementary Information Note S10 “COT calculation for the untethered BHMbot”.

Suggestion #4: What limits the speed? Why can't it move faster? What would allow faster motion? Is there a scaling argument?

Responses #4.1 for speed concern: Based on the analysis of the moving mechanism, the running speed of the BHMbot is equal to the produce of the bouncing frequency and bouncing length. For a BHMbot with a determined body length, there are three main influencing factors of the running speed, including the working current (frequency and amplitude), structural parameters and payload mass.

(1) Working current. Since the basic working principle of the electromagnetic actuator is forced vibration, the BHMbot can achieve the maximum running speed when the current frequency is close to its resonant frequency, as shown in Fig. R3a. When the current frequency deviates from the resonant frequency, the running speed drops gradually. The current amplitude determines the output power of the electromagnetic actuator, which also influences the speed of the BHMbot. A larger current amplitude can result in a higher relative running speed (Fig. R3b). We have discussed this issue in the Section “Tethered locomotion tests and parameters optimization”.

Fig. R3. Influence of the alternating current (frequency and amplitude) on the running of the BHMbot. **a**, Measurements of the relative running speed of four prototypes with different body lengths (10, 15, 20 and 25 mm) versus the alternating current (AC) frequency, with the AC amplitude remaining 0.15 A. **b**, Measurements of the relative running speed of the above four prototypes versus AC amplitude, with the AC frequency remaining constant (360, 200, 160 and 140 Hz respectively).

(2) Structural parameters. There are several structural parameters that are related to the locomotion performance, including the initial relative distance between the hollow coil and the magnet z , the width of the cantilever w_c of the electromagnetic actuators, the initial body tilt angle θ_0 , and the length of rear legs l_r . We utilize the proposed dynamic model (Fig. R4a and R4b) to optimize the above structural parameters aiming at the BHMbot with different payloads. Taking the bare BHMbot as an example, Fig. R4c and R4d show the normalized running speed maps of the bare BHMbot versus the above four parameters. Based on the two maps, we can obtain the optimal parameters. Benefiting from the parameter optimization, the maximum running speed of the bare BHMbot rises to 33.3 BL/s (17.5 BL/s for the initial parameters), as shown in Fig. R4e. The maximum running speed of the BHMbot with a payload of 2000 mg rises to 25 BL/s (14.5 BL/s for the initial parameters), as shown in Fig. R4f. This issue is discussed in the Section “Tethered locomotion tests and parameters optimization”.

Fig. R4. Dynamical modeling and parameters optimization of the BHMbot. **a**, Simplified planar dynamical model of the BHMbot to analyze the motion characteristics of the BHMbot. **b**, Dimensions of the planar dynamic model, including four generalized coordinates used to establish the dynamic equations of the model. **c**, Simulation results of normalized relative linear running

speed on the paper substrate versus cantilever width w_c and the relative distance between the hollow coil and the magnet z . **d**, Simulation results of normalized relative linear running speed on the paper substrate versus body tilt angle θ_0 and the distance between the rear feet and the COM of the body in the vertical body direction l . **e**, Optical photo of the tethered BHMbot using optimized structural parameters (prototype #3) moving forward at a maximum speed of 50 cm/s (33.3 BL/s). **f**, Optical photo of the tethered BHMbot (prototype #4) moving forward with a relative speed of 25 BL/s when carrying a hexagonal nut (M8, 2000 mg), which is more than five times its body mass (370 mg).

(3) Payload mass. The payload mass is a crucial factor that determines the running speed of the microrobot. For the BHMbot, experimental and simulation results both demonstrate that the running speed keeps increasing until the payload mass exceeds a certain value (m_{op}), as shown in Fig. R5a. Such a result is quite different from other existing microrobots which face severe running speed degradation after carrying payloads. After analyzing the inner mechanism that results in the variation tendency, it is found that the complementary combination of high bouncing frequency and bouncing length contributes to the high-speed locomotion performance of the BHMbot after carrying payloads. Consequently, the untethered BHMbot can still achieve high-speed locomotion when carrying the onboard payloads (such as the power and control units) with a mass close to the m_{op} . The untethered BHMbot, which is integrated with the power and control units, can achieve a running speed of 17.5 BL/s (Fig. R5b) and a turning centripetal acceleration of 65.4 BL/s² (Fig. R5c and R5d). We discuss this issue in the Section “Tethered locomotion analysis after carrying payloads”.

Fig. R5. Locomotion performance of the BHMbot after carrying payloads. **a**, Comparison of the simulation and experimental results of the maximum running speed versus the payload mass for

prototype #2.2. **b**, Optical photos of the BHMbot running linearly on a paper surface with a maximum speed of 17.5 BL/s. The red dotted lines represent the locomotive displacements of the BHMbot in 1.0 s. **c**, **d**, Optical photos of the BHMbot achieving anticlockwise and clockwise turns on the paper surface with a maximum relative centripetal acceleration of 65.4 and 39.4 BL/s² respectively. The gray dotted lines represent the angular displacements of the BHMbot in 0.5 s.

Responses #4.2 for scaling concern: According to the experimental results and theoretical analysis, the size has a significant influence on the running speed of the BHMbot. We fabricate a series of prototypes with different body lengths to test their running speed under varying working current and payload mass (Fig. R6a-R6c). The prototypes with smaller body lengths tend to have higher resonant frequencies and faster relative running speeds near the resonant state. The larger body lengths can result in higher optimal payload masses (m_{op}) and are less sensitive to the increase of the payload mass. The results of theoretical analysis based on dynamical modeling also prove this conclusion (Fig. R6d). We add the experimental results about the size determination in the **Section “Tethered locomotion tests and parameters optimization”**. The theoretical analysis about the scaling effect is added in the **Section “Scaling effects analysis”**.

Fig. R6. Scaling effects of the BHMbot. **a**, Measurements of the relative running speed of four prototypes with different body lengths (10, 15, 20 and 25 mm) versus the current frequency, with the current amplitude remaining 0.15 A. **b**, Measurements of the relative running speed of the above

four prototypes versus current amplitude, with the frequency remaining constant (360, 200, 160 and 140 Hz respectively). **c**, Measurements of the maximum relative running speed of the above four prototypes versus the payload mass near the resonant state, with the current amplitude remaining 0.15 A. **d**, Simulation results of the scaling performance of the BHMbot (the maximum relative running speed under varying payload mass), with the body length ranging from 10 mm to 25 mm. In the scaling process, it is assumed that all the geometric parameters of the components of the BHMbot are scaled down based on the ratio of the body length ϕ .

Suggestion #5: Why does it move fast? How can you prove the bouncing helps? How were the linkages designed. Does the friction of the feet matter?

Responses #5.1 for high speed: As mentioned in Responses #4.1, the fast locomotion performance of the BHMbot is related to its actuation mechanism and moving mechanism. Firstly, the locomotion of the BHMbot through periodical bouncing movements is similar to the running gait observed in some running mammals. The speed of the BHMbot is determined by the product of the bouncing frequency and bouncing length. Due to the high stiffness of the actuation mechanism, the bare BHMbot (15-mm-long) can be driven at a relatively high frequency (200-300 Hz) to reach its resonant state. It means that the BHMbot can go through hundreds of bouncing movements in one second, which leads to a high running speed. Secondly, the running speed of the BHMbot keeps increasing until the payload mass exceeds a certain value (m_{op}) due to the complementary combination of bouncing frequency and bouncing length. Consequently, the untethered BHMbot can still achieve high-speed locomotion when carrying the onboard payloads (such as the power and control units) with a mass close to m_{op} . The above issues have been discussed in detail in the **Section “Tethered locomotion tests and parameters optimization”** and the **Section “Tethered locomotion analysis after carrying payloads”**.

Responses #5.2 for bounce concern: Based on the videos captured by the high-speed camera, the forward locomotion is composed of a series of bouncing movements, during which the BHMbot demonstrates an apparent aerial phase with all legs in the air. During the aerial phase of the bouncing movements, the BHMbot can rush forward in

the air without the limitation of friction force, which results in a large bouncing length and running speed. The slow motion of the bouncing movements of the BHMbot during forward locomotion is presented in Supplementary Movie S2.

Responses #5.3 for linkage design: The design of the linkage is inspired by a four-bar linkage transmission used in micro flapping vehicles⁶, which is composed of rigid linkages and flexible hinges, as shown in Fig. R7. One end of the transmission (linkage L_3) is connected to the magnet and the other (linkage L_1) is connected to the support frames. Then, the reciprocating motion of the magnet can be transformed into the swing motion of the front leg through the transmission. To reduce the assembly error, the front leg is integrated with the transmission. This issue is discussed in the **Supplementary Information Note S3 “Design of the transmission”**.

Fig. R7. Enlarged top view of the actuation mechanism composed of the electromagnetic actuator, transmission and the front leg. The reciprocating linear motion of the electromagnetic actuators is transformed into the swing motion of the front legs.

Responses #5.4 for friction effects: The experimental and simulation results both indicate that the friction of the feet is related to the locomotion performance of the BHMbot. We have tested the locomotion performance of the untethered BHMbot on four surfaces with different levels of roughness (#1-#4, representing glass, wood, paper, and plastic respectively). The BHMbot can run forward on the above four surfaces with the maximum speeds of 4.4 BL/s, 7.5 BL/s, 17.5 BL/s, and 12.0 BL/s respectively (Fig. R8a). Fig. R8b shows the variation of the relative running speed of the untethered BHMbot versus the friction coefficient between the feet and the ground, which indicates that the running speed increases firstly and then declines as the friction coefficient rises.

The turning performance of the untethered BHMbot on the above four surfaces is shown in Fig. R8c, which exhibits a similar variation tendency. The BHMbot achieves clockwise turns on the four surfaces with the maximum relative turning centripetal accelerations of 19.7 BL/s^2 , 25.7 BL/s^2 , 39.4 BL/s^2 , and 38.55 BL/s^2 respectively. Fig. R8d-R8f shows the locomotion process of the BHMbot on the paper surface with the maximum relative running speed (17.5 BL/s) and turning centripetal accelerations (39.4 BL/s^2 for the clockwise turn and 65.4 BL/s^2 for the anticlockwise turn).

The locomotion performance of the untethered BHMbot on four substrates is discussed in the Section “Parameter optimization and untethered performance evaluation” in the last submission. In the new version, this part is in the Section “Untethered locomotion performance evaluation”.

Fig. R8. Influence of the friction on the running speed of the BHMbot. **a**, Maximum relative running speeds of the BHMbot on four different surfaces with different friction coefficients μ , numbered from #1 to #4, corresponding to glass, wooden, paper, and plastic surfaces. **b**, Variation of the maximum running speed of the BHMbot versus the friction coefficient between the feet and the ground. **c**, Maximum centripetal accelerations of the BHMbot during clockwise and anticlockwise turns on the above four surfaces. **d**, Optical photos of the BHMbot running linearly on a paper surface with a maximum speed of 17.5 BL/s . The red dotted lines represent the locomotive displacements of the BHMbot in 1.0 s . **e**, **f**, Optical photos of the BHMbot achieving

anticlockwise and clockwise turns on the paper surface with maximum relative centripetal accelerations of 65.4 and 39.4 BL/s² respectively. The gray dotted lines represent the angular displacements of the BHMbot in 0.5 s.

Suggestion #6: What performance on a slope?

Responses #6: We test the locomotion performance of the 15-mm BHMbot (prototype #3) on a series of slopes (0° - 7.2°), as shown in Fig. R9a. The tethered BHMbot can achieve an average speed of 6.5 BL/s while climbing a slope of 6°, as shown in Fig. R9b. In the last submission, we have discussed the issue in the **Method Section “Parameters optimization design”**. In the new version, we transfer this part to the **Section “Tethered locomotion tests and parameters optimization”**.

Fig. R9. Locomotion performance on slopes for the BHMbot. a, Experimental and simulation results of the relative running speed of the optimized BHMbot (prototype #3) versus the slope angle (from 0° - 7.2°). **b,** Optical photo showing prototype #3 moving on a slope of 6° with a maximum speed of 6.5 BL/s.

Suggestion #7: How much power is needed? I can't find a number in mW. How was COT computed?

Responses #7.1 for power concern: The power consumption of the BHMbot can be calculated based on the measured working voltage and current. For the untethered BHMbot (prototype #6), a galvanometer (UNI-T@ UT803) is connected to the circuit of the battery to measure the effective value of the input current I_b (0.48 A). Considering that the normal voltage of the used lithium battery is 3.7 V, the consumed power of the untethered BHMbot is confirmed as 1.77 W. For the tethered BHMbot (prototype #2.2), both the current and voltage signals are sinusoidal waves, and the phase difference between the two signals is about 0. Consequently, the power consumption could be

estimated as the product of the effective values of the voltage and current signals (1.38 V and 0.15 A), and the calculation result is 413.6 mW.

The power consumption of the untethered BHMbot has been discussed in the **Supplementary Information Note S5 “COT calculation for the untethered BHMbot” in the last submission**. We have transferred this part to the **Section “Cost of transport”** of the new version. The power consumption of the tethered prototype is added in **the Section “Tethered locomotion tests and parameters optimization”**.

Responses #7.2 for COT calculation: In this work, we select two COTs (COT_T and COT_M) to estimate the efficiency of the whole microrobot and the actuation mechanism for running gait respectively. Fig. R10 shows the power flow of an untethered BHMbot. The calculation of COT_T concentrates on the power flow from the battery to the microrobot, and the power P is equal to the output power of the battery P_b . For prototype #6, the measured P_b is 1.77 W and the value of COT_T is 303.7. To evaluate the energy efficiency of the actuation mechanism, we also calculate COT_M which only takes into account the power flow from the actuators to the microrobot, and P is equal to the output power of the electromagnetic actuators P_a . The calculated P_a is 5.62 mW, and the value of COT_M is 9.3, which indicates the high efficiency of the actuation mechanism.

The calculation of these two COTs is discussed in the **Supplementary Information Note S5 “COT calculation for the untethered BHMbot”** in the last submission. We transferred this part to the **Section “Cost of transport”** in the new version. More details are supplemented in **Supplementary Information Note S10 “COT calculation for the untethered BHMbot”**.

Fig. R10. Power flow from the battery to the locomotion of the untethered BHMbot.

Suggestion #8: It should be made clearer how this is different from the authors earlier work (ref 11), images look similar to this work.

Responses #8: There are significant differences between the current work and the earlier work, including the actuation mechanism, locomotion gait and control strategy.

(1) In this work, we present a new actuation mechanism to realize a high-speed running gait in a 2-cm microrobot, which is composed of two independent actuators, two four-bar linkage transmission mechanisms and two rigid actuated legs. The actuation mechanism of the robot presented in the earlier work (ref 11) is composed of only one actuator and two elastic legs (without transmissions), which is different from the actuation mechanism adopted by BHMbot.

(2) The locomotion gaits or moving mechanisms applied by the two microrobots are also different. As mentioned in the response to the major concern #1, the locomotion of the BHMbot through periodical bouncing movements is similar to the running gaits of several mammals (such as cheetahs). Furthermore, the speed of the BHMbot keeps increasing until the payload mass exceeds a certain value (marked as m_{op}) benefiting from the bioinspired running gait. Therefore, the untethered BHMbot can still achieve high-speed locomotion with the mass of the onboard payloads (power and control units) close to the m_{op} . As a comparison, the locomotion of the microrobot in the earlier work is composed of squirm cycles without apparent bouncing movements, which is similar to the locomotion gaits of several mollusks (such as inchworms). The forward movement is realized by the elastic deformation of the simply supported beams and the front legs, which is highly sensitive to the added payload mass. In this case, the microrobot in the earlier work will encounter inevitable speed degradation after carrying payloads due to the restricted deformation of elastic components (decrease from 18.9 BL/s to 4 BL/s after carrying a payload mass of 800 mg), which is similar to the variation trend of the soft microrobots.

(3) The control strategies of two microrobots are different. Due to the limitation of the single actuator, the earlier microrobot can only achieve straight locomotion by adjusting the body structure before testing, and the direction control can not be realized by using this configuration. Instead, the BHMbot can achieve the control of the motion direction and trajectories by adjusting the working frequency of two independent

actuators. Based on the control strategy, we design a micro power and control circuit (10 mm × 20 mm, 600 mg) and demonstrate high-speed untethered locomotion and trajectory control after the integration of the BHMbot and the above circuit, which is not achieved by the earlier work.

Suggestion #9: Not enough info on how sizes were determined.

Responses #9: Thanks for your suggestion. As mentioned in Responses #4.2 for scaling effects, we fabricate a series of prototypes with different body lengths to test their running speed under varying payload mass. Experimental results indicate that a smaller size can result in a larger relative running speed, but poorer load-carrying capacity (Fig. R6a-R6c). Considering that the estimated mass of the power and control units for generating an alternating current signal is about 1.4g, we select 15 mm as the body length of the BHMbot to achieve a faster untethered running speed. The theoretical analysis of the scaling effect based on the dynamical modeling is also conducted (Fig. R6d).

The experimental content related to size determination is added in the Section “Tethered locomotion tests and parameters optimization”, and the theoretical analysis of scaling effects is added in the Section “Scaling effects analysis”.

Fig. R6. Scaling effects of the BHMbot. a, Measurements of the relative running speed of four

prototypes with different body lengths (10, 15, 20 and 25 mm) versus the current frequency, with the current amplitude remaining 0.15 A. **b**, Measurements of the relative running speed of the above four prototypes versus current amplitude, with the frequency remaining constant (360, 200, 160 and 140 Hz respectively). **c**, Measurements of the maximum relative running speed of the above four prototypes versus the payload mass near the resonant state, with the current amplitude remaining 0.15 A. **d**, Simulation results of the scaling performance of the BHMbot (the maximum relative running speed under varying payload mass), with the body length ranging from 10 mm to 25 mm. In the scaling process, it is assumed that all the geometric parameters of the components of the BHMbot are scaled down based on the ratio of the body length ϕ .

Reference

- 1 Zou, Y., Zhang, W. & Zhang, Z. Liftoff of an Electromagnetically Driven Insect-Inspired Flapping-Wing Robot. *IEEE Trans. Rob.* **32**, 1285-1289, doi:10.1109/TRO.2016.2593449 (2016).
- 2 Mao, G. *et al.* Ultrafast small-scale soft electromagnetic robots. *Nat. Commun.* **13**, 4456 (2022).
- 3 Liang, J. *et al.* Electrostatic footpads enable agile insect-scale soft robots with trajectory control. *Sci. Rob.* **6**, eabe7906 (2021).
- 4 Goldberg, B. *et al.* Power and Control Autonomy for High-Speed Locomotion With an Insect-Scale Legged Robot. *IEEE Rob. Autom. Lett.* **3**, 987-993 (2018).
- 5 Jayaram, K., Shum, J., Castellanos, S., Helbling, E. F. & Wood, R. J. Scaling down an insect-size microrobot, HAMR-VI into HAMR-Jr. in *2020 IEEE International Conference on Robotics and Automation (ICRA)*. 10305-10311.
- 6 Wood, R. J. Liftoff of a 60mg flapping-wing MAV. in *2007 IEEE/RSJ International Conference on Intelligent Robots and Systems*. 1889-1894.

REVIEWER COMMENTS

Reviewer #1 (Remarks to the Author):

In my assessment, the authors thoroughly addressed all the comments and concerns in my original review.

I believe that this paper is excellent and worth of publication.

Reviewer #1 (Remarks on code availability):

All the relevant information is in the main manuscript and movies. I did not run any code.

Reviewer #2 (Remarks to the Author):

The manuscript is improved. It will be of interest to the robotics community.

i recommend accepting proving the power used is reported in the abstract and the bio-inspiration aspect addressed.

I do not think the authors understand the meaning of bioinspiration. In the revised manuscript, the authors did not change their device, but changed the "inspiration" from a grasshopper to a cheetah! Thus unwittingly demonstrating the absence of bioinspiration in their work...

Reviewer #3 (Remarks to the Author):

The authors report on BHMbot, a 2 cm long untethered legged robot weighing 1760 mg that is capable at moving up to 17.5 BL/s. The robot's speed and trajectory are relatively well controlled, and the overall untethered robot performance is very impressive. Below, I'll outline a few recommendations.

Recommendation 1: Avoid making remarks like "fastest speed ever reported". The abstract sites that the BHMbot is the fastest speed ever reported among insect-scale legged microrobots to the best of their knowledge, but citation 3 "Moving Mechanism of a High-speed Insect-scale Microrobot via Electromagnetically Induced Vibration" reports in the abstract that the fastest speed amongst published insect-scale microrobots is $20\text{cm}\cdot\text{s}^{-1}$ or 20 body lengths per second. If it was meant that this is the fastest untethered legged insect-scale microrobot less than 2cm then that should be specified in the abstract, similar to how the introduction specifies "the highest untethered relative running speed ... ever reported among untethered insect-scale legged microrobots under the dimension limitation of 2 cm".

Recommendation 2: Introduction sentence 1, the use of "pushes" is a bit odd and would be better replaced with "inspires" or "motivates".

Recommendation 3: The claims in the first paragraph and the rest of the paper should be updated to reflect the following untethered legged robots that are all less than 5cm in length:

1. "Miniature Autonomous Robot Based on Legged In-Plane Piezoelectric Resonators with Onboard Power and Control". 2022 Oct 24. DOI: 10.3390/mi13111815
2. "Power Autonomy and Agility Control of an Untethered Insect-Scale Soft Robot". 9 Aug 2023. DOI: <https://doi.org/10.1089/soro.2021.0201>

Recommendation 4: It is not clear if the walking vs running gait of the BHMbot was purposefully designed. How did the swing forth vs swing back modes of locomotion come about, was this directly inspired by the running gait of insects, was it inspired by other legged robots, or was it just the easiest movement to manufacture? An additional sentence explaining the origin of the design, as well as a citation of additional prior works that investigated various gaits for insect-scale legged robots should be included : <https://doi.org/10.1002/aisy.202300181>

Recommendation 5: For Figure 2.g and Figure 2.h. I recommend using 1 color gradient for both subfigures to declutter the overall figure, specifically the one on Fig 2.h that has 2 significant digits for each value in the gradient, unlike Fig 2.g.

Recommendation 6: The sound detection is a nice sensory addition. Using onboard sensors for autonomous locomotion, even if in a simple Braitenberg vehicle demonstration, would make the work

even stronger though. However, this is not required as I believe the work to already be impactful enough for publication as is.

Recommendation 7: The Australian tiger beetle should be added to Fig. 5 and Supplementary Table 10, as it can reach a speed of 2.5 m/s or 171 body lengths per second. Also, tiny Californian mites can run nearly 200 times their body lengths per second and should be added for scale as well. The claims in the introduction should be adjusted within the context of these insects.

Recommendation 8: Fig. 5f includes a wheeled robot in the “reported untethered legged robots” purple diamond region on the graph. This should either be removed, or a comparison to the MilliMobile robot should be added to Fig. 5f and Supplementary Table 10 as it had a better performance - Body Mass (g) 1.1g and BL/s 0.55: <https://dl.acm.org/doi/10.1145/3570361.3613304>

Recommendation 9: Should add citation on line 519 for micro cameras that have been designed at the insect scale: <https://doi.org/10.1126/scirobotics.abb0839>

Recommendation 10: A relative error column to Supplementary Table 3, similar to the relative error row in Supplementary Table 2, would be beneficial.

Reviewer #3 (Remarks on code availability):

The code is only available upon request from the corresponding authors, as such I did not access it in doing this review to remain anonymous during the review process.

Point-by-point Response to Reviewers' Comments

Dear reviewers,

We have received the reviewers' comments regarding our manuscript "**A Wireless Controlled Robotic Insect with Ultrafast Untethered Running Speed Based on Bioinspired Running Gait**" (NCOMMS-23-31631B) in the last submission and those comments are all valuable and helpful in improving our manuscript. The authors express their sincere gratitude to the reviewers for their efforts in reviewing the manuscript.

In this submission, we have considered all the comments carefully and addressed all of them in the revised manuscript entitled "**A Wireless Controlled Robotic Insect with Ultrafast Untethered Running Speeds**". The point-by-point response letter is also attached below to address the issues presented by the reviewers. Before a point-by-point response, we want to briefly summarize the responses to the reviewers' comments.

To Reviewer #1: The authors sincerely thank the reviewer for the recommendation to our work. Two key codes for the design of the BHMbot are submitted in this version.

To Reviewer #2: The authors sincerely thank the reviewer for the insightful comments. We have accepted the reviewer's suggestions and deleted all the statements of bioinspiration throughout the manuscript and the title of the manuscript is revised as "**A Wireless Controlled Robotic Insect with Ultrafast Untethered Running Speeds**". The detailed calculation process of the consumed power (1.77 W) of the untethered BHMbot is highlighted in the Supplementary Information and this response letter.

To Reviewer #3: The authors sincerely thank the reviewer for the good comments. We have extended literature studies as suggested by the reviewer. The design origin of the BHMbot is also explained in the main text and the response letter. The remarks such as "novel", "highest", and "fastest" are deleted in the revised manuscript.

The authors sincerely thank all the reviewers for their valuable suggestions. We hope the revised manuscript can meet the publication criteria of **Nature Communications**. We appreciate your time and look forward to your responses again.

The point-by-point responses to three reviewers are as follows.

Point-by-point Response to Reviewer #1

1.1 Response to the main comments

Comment:

In my assessment, the authors thoroughly addressed all the comments and concerns in my original review. I believe that this paper is excellent and worth of publication.

Response:

The authors sincerely thank the reviewer for the recommendation to our work.

1.2 Response to the detailed suggestions

Suggestion #1:

All the relevant information is in the main manuscript and movies. I did not run any code.

Responses #1:

In this work, **the most important codes include (1) code 1 for dynamic model and (2) code 2 for remote control.** The first code is used to solve the dynamic equations of the BHMbot to obtain its locomotion performance parameters when carrying various payloads. The second code is used to ensure the operation of the power and control circuit, and realize the direction and trajectory control of the BHMbot via the remote control of mobile phones.

In this new version, we have submitted the two aforementioned codes for review. Two text files have been submitted to offer detailed explanations of the respective codes.

The authors sincerely thank the reviewer for the good comments.

Point-by-point Response to Reviewer #2

2.1 Response to the main comments

Comment:

The manuscript is improved. It will be of interest to the robotics community. I recommend accepting providing the power used is reported in the abstract and the bio-inspiration aspect addressed.

Response:

We are very grateful for your recommendation and insightful comments. Based on your comments, we have revised the manuscript accordingly and addressed them in the following response.

2.2 Response to the detailed suggestions

Suggestion #1:

Providing the power used is reported in the abstract.

Responses #1:

The consumed power P_b of the untethered BHMbot can be expressed as:

$$P_b = V_b I_b \quad (1)$$

where V_b is the rated voltage of the battery (3.7 V), and I_b is the effective value of the output current of the battery during the running movement of the untethered BHMbot. Fig. R1 shows the equivalent circuit diagram of the untethered BHMbot. A galvanometer (UNI-T[®] UT803) is connected to the circuit in series to measure I_b . Considering the influence of the resistance of the extra wire connecting the galvanometer and the battery (R_w), we record the measurements of I_b when three segments of wire with different lengths ($R_w = 0.3 \Omega$, 0.7Ω , and 1.2Ω) are connected to the circuit. The measured I_b is 0.47 A, 0.46 A, and 0.45 A, respectively (the frequencies of the two driving channels are both 220 Hz). The relationship between I_b and R_w can be approximately expressed as:

$$I_b = 0.0056R_w^2 - 0.0306R_w + 0.4787 \quad (2)$$

When R_w drops to zero, an estimated value for I_b is 0.4787 A. Therefore, the consumed power P_b is 1.77 W. The relatively large power consumption is mainly caused by the heating effect of the electromagnetic actuator and the onboard circuit.

The power consumption is discussed in Supplementary Note S10 “COT calculation for the untethered BHMbot” in the last submission (Supplementary Information, Page 15).

Fig. R1. Equivalent circuit diagram of the BHMbot. V_b represents the output voltage of the battery; I_b represents the output current of the battery; R_w represents the resistance of the wire connecting the battery and the galvanometer.

Suggestion #2:

I do not think the authors understand the meaning of bioinspiration. In the revised manuscript, the authors did not change their device, but changed the “inspiration” from a grasshopper to a cheetah! Thus unwittingly demonstrating the absence of bioinspiration in their work...

Responses #2:

Thanks for your professional comment, and we have accepted your suggestion. The authors apologize for the improper statements about “bioinspiration” in the last two submissions. **In essence, only the running gait or the bouncing movement of the BHMbot is similar to the running postures of several mammals, while the actuation mechanism of the legs is different from the musculoskeletal systems of mammals.** As the reviewer pointed out, it is not proper to emphasize the term of “bioinspiration”.

In this new version, we have deleted the statements related to “inspiration” and “bioinspiration” (line 19, line 63, and line 66). The title of the manuscript is also

revised to “A Wireless Controlled Robotic Insect with Ultrafast Untethered Running Speeds” (“Bioinspired running gait” is removed). The natural mammals and insects are mentioned only to compare the locomotion gait and performance.

For the change of the “inspiration” from a grasshopper to a cheetah in the last two submissions, we would like to offer a detailed explanation.

In the initial submission, based on the moving mechanism analysis, the BHMbot utilizes leg actuation to drive the whole body to move forward with an apparent aerial phase. Considering that the grasshopper utilizes leg actuation to jump into the air and is similar in size to the BHMbot, it is selected as the bioinspiration template. As the reviewer pointed out in the first round of review, grasshopper excels at jumping and can achieve a jumping height of several body heights, which is impossible for the BHMbot. Therefore, it is indeed improper to select grasshoppers as a bioinspiration template.

In the last submission, according to the reviewer’s suggestion, we carefully reanalyze the locomotion process of the BHMbot via high-speed videos. It is found that the forward locomotion of the BHMbot near the resonant state is composed of periodical bouncing movements with apparent aerial phases. During one bouncing cycle, the BHMbot bounces off the ground and ascends obliquely, and then lands on the ground under the action of its gravity. The locomotion process is similar to several existing running microrobots^{1,2}. Referring to the literature², it is believed that the locomotion gaits of the BHMbot and the SEMR are similar to those of several running mammals, such as the cheetah. Therefore, we select the cheetah as the bioinspiration template in the last submission. From the perspective of design origin, the actuation mechanism of BHMbot is different from the musculoskeletal systems of cheetahs. In this case, it is also improper to use the term of “bioinspiration”.

The authors sincerely thank the reviewer for the effort in reviewing and improving the manuscript.

2.3 Reference

- 1 Wu, Y. *et al.* Insect-scale fast moving and ultrarobust soft robot. *Sci. Rob.* **4**, eaax1594, (2019).
- 2 Mao, G. *et al.* Ultrafast small-scale soft electromagnetic robots. *Nat. Commun.* **13**, 4456, (2022).

Point-by-point Response to Reviewer #3

3.1 Response to the main comments

Comment:

The authors report on BHMbot, a 2 cm long untethered legged robot weighing 1760 mg that is capable of moving up to 17.5 BL/s. The robot's speed and trajectory are relatively well controlled, and the overall untethered robot performance is very impressive. Below, I'll outline a few recommendations.

Response:

We are very grateful for your insightful suggestions, and these suggestions are very helpful in both revising this manuscript and guiding our research in the future. According to your suggestions, we have revised the manuscript accordingly.

3.2 Response to the detailed suggestion

Suggestion #1:

Avoid making remarks like “fastest speed ever reported”. The abstract sites that the BHMbot is the fastest speed ever reported among insect-scale legged microrobots to the best of their knowledge, but citation 3 “Moving Mechanism of a High-speed Insect-scale Microrobot via Electromagnetically Induced Vibration” reports in the abstract that the fastest speed amongst published insect-scale microrobots is $20\text{cm}\cdot\text{s}^{-1}$ or 20 body lengths per second. If it was meant that this is the fastest untethered legged insect-scale microrobot less than 2cm then that should be specified in the abstract, similar to how the introduction specifies “the highest untethered relative running speed ... ever reported among untethered insect-scale legged microrobots under the dimension limitation of 2 cm”.

Responses #1:

Thanks for your suggestion and we have accepted it. It is noted that the fastest speed of 20 body lengths per second (20 BL/s) referred in citation 3 is achieved by a tethered microrobot connecting with an external power source¹. When integrated with an

onboard battery and a power electronics circuit³, the speed of the microrobot drops to 1.2 BL/s.

We have deleted all similar remarks in the new version, such as “fastest” and “highest”, to follow a sober writing style (line 27, line 67, and line 437).

Suggestion #2:

Introduction sentence 1, the use of “pushes” is a bit odd and would be better replaced with “inspires” or “motivates”.

Responses #2:

We have accepted your suggestion. As suggested, we have replaced “pushes” with “motivate” (line 36).

Suggestion #3:

The claims in the first paragraph and the rest of the paper should be updated to reflect the following untethered legged robots that are all less than 5cm in length:

1. “Miniature Autonomous Robot Based on Legged In-Plane Piezoelectric Resonators with Onboard Power and Control”. 2022 Oct 24. DOI: 10.3390/mi13111815
2. “Power Autonomy and Agility Control of an Untethered Insect-Scale Soft Robot”. 9 Aug 2023. DOI: <https://doi.org/10.1089/soro.2021.0201>

Responses #3:

We have accepted your suggestion. As suggested, we have updated relevant claims in the first paragraph (line 42) and the rest of the manuscript to reflect the above untethered legged robots. Fig. 5f (line 412) and Supplementary Table 10 (Supplementary Information, Page 32) have also been updated to include these two robots.

Suggestion #4:

It is not clear if the walking vs running gait of the BHMbot was purposefully designed. How did the swing forth vs swing back modes of locomotion come about, was this directly inspired by the running gait of insects, was it inspired by other legged robots, or was it just the easiest movement to manufacture? An additional sentence explaining the origin of the design, as well as a citation of additional prior works that investigated various gaits for insect-scale legged robots should be included: <https://doi.org/10.1002/aisy.202300181>

Responses #4:

Based on the literature research^{4,5}, the running gait can achieve larger stride frequency and length, which proposes an effective way to realize relatively high moving speeds for insect-scale microrobots^{6,7}. **To achieve high locomotion agility comparable to natural insects, the BHMbot in this work is purposefully designed to operate in a running gait similar to that of several mammals and insects.**

To demonstrate the running gait, a microrobot should go through an apparent aerial phase with all legs suspended in the air during the locomotion, which requires a bouncing momentum from the ground. The reported running microrobots tend to utilize body deformation to obtain the bouncing momentum, and then achieve periodical bouncing movements of the body at high frequency to mimic the running postures of several mammals^{1,2}. However, after integrating with power and control units, untethered running speeds of these microrobots decrease dramatically since the body deformation induced bouncing movement is directly affected by the added payload.

Apart from utilizing body deformation at high frequency^{1,2}, the bouncing momentum can also be generated by the impacts between swinging rigid legs and the ground. **Since the direction of the bouncing momentum is obliquely upward, the swing motion of legs should be in an obliquely downward direction when they impact the ground. This specific swing motion of legs forms the fundamental design principle of the BHMbot.** In this case, the swing motions of legs utilized by the BHMbot are purposefully designed based on the kinematics analysis of running gait.

We have added a paragraph (**line 75**) to explain the origin of the design of swing

modes in **Section “Design and moving mechanism”**. The mentioned literature about the investigation of various gaits for insect-scale legged robots has also been cited (**line 76**).

Suggestion #5:

For Figure 2.g and Figure 2.h. I recommend using 1 color gradient for both subfigures to declutter the overall figure, specifically the one on Fig 2.h that has 2 significant digits for each value in the gradient, unlike Fig 2.g.

Responses #5:

We have accepted your suggestion. As suggested, one common color gradient for both Figure 2.g and Figure 2.h is used in the new version (**line 163**).

Suggestion #6:

The sound detection is a nice sensory addition. Using onboard sensors for autonomous locomotion, even if in a simple Braitenberg vehicle demonstration, would make the work even stronger though. However, this is not required as I believe the work to already be impactful enough for publication as is.

Responses #6:

Thanks for your professional suggestion and recommendation. The autonomous obstacle avoidance based on an insect-scale visual system is a significant next step for the BHMbot, and our research in this area has spanned approximately a year. However, it is a pity that additional time is still required to accomplish the circuit design, programming design, and autonomous obstacle avoidance demonstration.

Considering the time constraint of the revision, the autonomous locomotion demonstration is not added to the revision. We consider the remote control demonstrated in this work as the initial milestone for the BHMbot and the next milestone for further exploration will be the autonomous locomotion of the BHMbot.

Suggestion #7:

The Australian tiger beetle should be added to Fig. 5 and Supplementary Table 10, as it can reach a speed of 2.5 m/s or 171 body lengths per second. Also, tiny Californian mites can run nearly 200 times their body lengths per second and should be added for scale as well. The claims in the introduction should be adjusted within the context of these insects.

Responses #7:

We have accepted your suggestion. As suggested, we have added the Australian tiger beetle (*Cicindela eburneola*⁸) and the Californian mite (*Paratarsotomus macropalpis*⁹) to Fig. 5 (**line 412**) and Supplementary Table 10 (**Supplementary Information, Page 32**). The claims in the introduction have also been adjusted within the context of these insects (**line 34**).

Suggestion #8:

Fig. 5f includes a wheeled robot in the “reported untethered legged robots” purple diamond region on the graph. This should either be removed, or a comparison to the MilliMobile robot should be added to Fig. 5f and Supplementary Table 10 as it had a better performance - Body Mass (g) 1.1g and BL/s 0.55: <https://dl.acm.org/doi/10.1145/3570361.3613304>.

Responses #8:

We have accepted your suggestion. As suggested, we have added a comparison to the MilliMobile robot in Fig. 5f (**line 412**) and Supplementary Table 10 (**Supplementary Information, Page 32**), and we have removed the restriction of “legged”.

Suggestion #9:

Should add citation on line 519 for micro cameras that have been designed at the insect scale: <https://doi.org/10.1126/scirobotics.abb0839>.

Responses #9:

We have accepted your suggestion. As suggested, we have added the citation¹⁰ in the new version (line 530).

Suggestion #10:

A relative error column to Supplementary Table 3, similar to the relative error row in Supplementary Table 2, would be beneficial.

Responses #10:

We have accepted your suggestion. As suggested, we have added a relative error column to Supplementary Table 3 in the new version (Supplementary Information, Page 28).

Suggestion #11:

The code is only available upon request from the corresponding authors, as such I did not access it in doing this review to remain anonymous during the review process.

Responses #11:

In this work, the most important codes include (1) code 1 for dynamic model and (2) code 2 for remote control. The first code is used to solve the dynamic equations of the BHMbot to obtain its locomotion performance parameters when carrying various payloads. The second code is used to ensure the operation of the power and control circuit, and realize the direction and trajectory control of the BHMbot via the remote control of mobile phones.

In this new version, we have submitted the two aforementioned codes for review. Two text files have been submitted to offer detailed explanations of the respective codes.

The authors sincerely thank the reviewer for the good comments.

3.3 Reference

- 1 Wu, Y. *et al.* Insect-scale fast moving and ultrarobust soft robot. *Sci. Rob.* **4**, eaax1594, (2019).
- 2 Mao, G. *et al.* Ultrafast small-scale soft electromagnetic robots. *Nat. Commun.* **13**, 4456, (2022).
- 3 Liang, J. *et al.* Electrostatic footpads enable agile insect-scale soft robots with trajectory control. *Sci. Rob.* **6**, eabe7906, (2021).
- 4 Michaels, T. M. MECHANICS OF A RAPID RUNNING INSECT : TWO-, FOUR-AND SEX-

LEGGED LOCOMOTION.

- 5 Alexander, R. M. Walking and running. *The Mathematical Gazette* **80**, 262-266, (1996).
- 6 Kabutz, H. & Jayaram, K. Design of CLARI: A Miniature Modular Origami Passive Shape-Morphing Robot. *Advanced Intelligent Systems* **5**, 2300181, (2023).
- 7 Baisch, A. T., Ozcan, O., Goldberg, B., Ithier, D. & Wood, R. J. High speed locomotion for a quadrupedal microrobot. *The International Journal of Robotics Research* **33**, 1063-1082, (2014).
- 8 Kamoun, S. & Hogenhout, S. A. Flightlessness and Rapid Terrestrial Locomotion in Tiger Beetles of the Cicindela L. Subgenus Rivacindela van Nidek from Saline Habitats of Australia (Coleoptera: Cicindelidae). *The Coleopterists Bulletin* **50**, 221-230, (1996).
- 9 Rubin, S., Young, M. H.-Y., Wright, J. C., Whitaker, D. L. & Ahn, A. N. Exceptional running and turning performance in a mite. *J. Exp. Biol.* **219**, 676-685, (2016).
- 10 Iyer, V., Najafi, A., James, J., Fuller, S. & Gollakota, S. Wireless steerable vision for live insects and insect-scale robots. *Sci. Rob.* **5**, eabb0839, (2020).

REVIEWERS' COMMENTS

Reviewer #3 (Remarks to the Author):

The authors properly and thoroughly addressed all the feedback in my and the other reviewer's original reviews, therefore I believe this paper is ready for publication in Nature Communications.

Reviewer #3 (Remarks on code availability):

The relevant information was provided in the main manuscript and supplementary files, therefore I did not run any code.